# Searching for actual causes:
# Approximate algorithms with adjustable precision

## Abstract

Causality has gained increasing attention in recent years, notably for improving the interpretability of machine learning models. Yet the field of explainable artificial intelligence (XAI) has been criticized for emphasizing general tendencies rather than the situation-specific facts, which users typically expect as explanations. These expectations align with the notion of actual causes, which identify what made the observed outcome happen, in the specific context at hand. Halpern and Pearl provided a formal basis for actual causation, but identifying actual causes is NP-complete. Practical identification algorithms are extremely scarce, restricted to narrow classes of models, and typically identify only the shortest cause. We address this gap between the formal theory and its applicability through two main contributions. First, we introduce a baseline approximate polynomial-time algorithm with adjustable precision, together with two complementary algorithms that improve its efficiency. Second, we provide a theoretical result showing that the actual-cause identification problem can be decomposed into smaller sub-instances that preserve the set of solutions. This result directly motivates one of the complementary algorithms. Our experiments demonstrate that the baseline method can approximate the set of actual causes, notably for non-Boolean and stochastic models, and that the complementary algorithms further improve its performance.

## 1 Introduction

Causality offers several critical advantages for machine learning by enabling models to generalize beyond correlations observed in training data. Causal models distinguish between associations and true cause-effect relationships, which allows robustness to distributional shifts and interventions (Schölkopf et al., 2021; 2012). Such models also support counterfactual reasoning, which is essential for reliable decision-making in dynamic environments (Pearl, 2009). Incorporating causal knowledge enhances sample efficiency (Castro et al., 2020), addressing one of the most important challenges in modern deep learning—namely, data scarcity (Alzubaidi et al., 2023). Finally, causal frameworks improve interpretability by aligning explanations with human-understandable causal mechanisms (Moraffah et al., 2020; Schwab & Karlen, 2019; Madumal et al., 2020). Significant research has been devoted to leveraging these strengths and identifying causal structures from data (Guo et al., 2020; Assaad et al., 2022).

Most of this work, however, concerns general, type-level causation (which factors bring about a kind of outcome in general) whereas explanation calls for actual, token-level causation, which identifies the facts that caused a particular outcome (Halpern, 2016; Hitchcock, 2007). This distinction is crucial because users typically desire specific and concrete explanations of the observed situation (Miller, 2019). For example, while the statement "Smoking causes cancer" provides a broad understanding of a causal tendency, a person diagnosed with cancer typically seeks a situation-specific explanation, such as "Your cancer was caused by your smoking."

---

Disclosure: A large language model was used only for language polishing and readability improvements in parts of this manuscript. All technical content, claims, and final wording were reviewed by the authors.

| Causation: Why did this happen? | | |
|---|---|---|
| | **General Causation** | **Actual Causation** |
| Formulation | What **factors** contributed to this? | What **facts** caused this? |
| Example | Smoking causes cancer. | Their smoking caused their cancer. |
| Definition | Bayesian networks
Mainly accepted
Large literature | HP-causes
Still under debate
Some literature |
| Identification | Causal discovery
No working solution
Huge literature | No unified field
No practical solution
Little literature |

Table 1: General causation versus Actual causation. This paper addresses the red area, i.e., identifying actual causes.

Such "token level" analysis is ubiquitous in Explainable Artificial Intelligence (XAI), notably via feature-attribution methods such as LIME (Ribeiro et al., 2016) and SHAP (Lundberg & Lee, 2017), which score how much each input influences a black-box model's prediction. As Miller (2019) argues, such a list of weighted contributions is not yet an explanation: an explanation must be contrastive and selective. These considerations hint toward the usage of actual causes. Beyond XAI, actual causes can be used to improve control over systems and mitigate unwanted behaviors (Reyd et al., 2025), as "canceling" the cause can prevent the consequence. Actual causes can also be used to assess degrees of responsibility (Halpern, 2015) or assign blame (Chockler & Halpern, 2004).

Intuitively, an actual cause is a fact that made the difference to an observed consequence: had the cause been absent the consequence would not have occurred. However, providing a satisfactory definition of actual causes has been a longstanding challenge in philosophy (Hume, 1739; Lewis, 1974), logic, and computer science (Halpern & Pearl, 2001; 2005a; Beckers, 2021). Halpern & Pearl (2005b) proposed the most influential formalism, often referred to as the HP definition or HP cause. Although it is widely used and regarded as a solid baseline, it still has limitations, such as its treatment of responsibility (Chockler & Halpern, 2004) and normality (Halpern & Hitchcock, 2015). It also falls short of producing satisfactory causal explanations, which must be contrastive and account for user expectations (Miller, 2019; 2021). However, the primary obstacle to adopting actual causation in XAI lies in its challenging identification and in the limited attention this task has received.

Identifying actual causes is NP-complete (Aleksandrowicz et al., 2017). Practical solutions for finding exact or approximate HP causes are scarce. Approaches that remain close to the definition are generally restricted to identifying single causes in Boolean systems and require knowledge of the system's equations (Ibrahim & Pretschner, 2020; Hopkins, 2002). Other approaches approximate alternative notions inspired by the HP definition (Albantakis et al., 2019; Reyd et al., 2024). Overall, to the best of our knowledge, no practical approach exists to identify the full set of HP causes, nor to identify any HP cause when the model is non-Boolean or stochastic. Table 1 summarizes this situation: while general causation is widely studied, actual causation, especially its identification, receives limited attention. This paper addresses this gap in the causality and XAI literature.

We provide two main contributions. (1) We propose a set of practical algorithms, with limited assumptions, polynomial-time complexity, and adjustable precision, for identifying the set of actual causes. This includes a baseline algorithm, the first to address full HP-cause identification while supporting non-Boolean and stochastic models, and two complementary algorithms that improve its efficiency. (2) We introduce and formally prove a theoretical result stating that the HP-cause identification task can be divided into sub-instances that preserve solutions, thereby easing the search process. One of our complementary algorithms is a direct implementation of this result.

We present three algorithms as our first contribution. The Minimal Beam Search (MBS) algorithm, our baseline, adapts Beam Search (Lowerre, 1976) and leverages the minimality requirement of the HP definition to explore counterfactual worlds and identify actual causes. It uses an oracle to determine whether the

consequence holds in a given counterfactual situation, along with a heuristic function that estimates how close the consequence is to being "canceled". The Iterative Sub-Instance (ISI) algorithm leverages our theoretical result and, when the system structure is available, decomposes the search space into smaller subspaces. The Lower Upper Confidence Bound (LUCB) algorithm improves the efficiency of estimating the oracle and heuristic functions in stochastic models. Our algorithms assume variables with discrete domains. They can be used through a Python module not linked here for anonymity.

The novelty of our approach lies in two aspects. First, to the best of our knowledge, we are the first to propose algorithms to identify actual causes in general SCM (instead of solely deterministic boolean ones) and to aim for the full set of causes (instead of only one instance). Second, our theoretical result and the algorithm they inspire (our second algorithm) are fully novel. The first and third algorithms are somewhat novel as we propose adaptations from the baselines of the original version. However, we do not claim that algorithms 1 and 3 are substantial novel theoretical contributions. We claim practical novelty in the approach to the problem and theoretical novelty in the second contribution.

We evaluate our methods on a Boolean system of varying size introduced by Ibrahim & Pretschner (2020), and extend it to non-Boolean and stochastic variants. Our results show that our baseline algorithm approximates the set of HP causes and enables an actionable trade-off between quality and efficiency. Our complementary algorithms consistently improve the baseline's performance. The experimental code is provided in a public repository, not linked for anonymity. An anonymized implementation accompanying this submission is provided in the supplementary material.

## 2   Background

An actual cause is defined as a counterfactual or but-for cause, i.e., but for the cause, the consequence would not have happened. In this section, we define the formalism of Structural Causal Models (SCM), necessary for counterfactual reasoning, the formal definition of actual causation by Halpern and Pearl, and some of its limitations. We illustrate these notions using the classic rock-throwing example.

**Example 1 (Rock throwing)** *Suzy and Billy are throwing rocks at a bottle. Both aim accurately and Suzy hits the bottle first. Therefore, the bottle shatters. In this scenario, we intuitively consider that the cause of the bottle shattering is Suzy's throw or Suzy's hit.*

### 2.1   Structural Causal Models

Pearl (2009) introduced SCMs to model counterfactual reasoning. In our example, we can model what would happen if Suzy did not hit. While still assuming she aimed correctly, we consider an imaginary world where, despite her accurate throw, she missed. An SCM consists of variables that model our observations of the system and of functions that set the variable values based on the system's dynamics. SCMs also include "exogenous variables" to initialize the endogenous variables' values. Each endogenous variable value is computed using the corresponding exogenous variable and a set of other endogenous variables called its "causal parents".

**Definition 1 (Structural Causal Model)** *We denote an SCM $\mathcal{M} = (\mathcal{V}, \mathcal{U}, \mathrm{F}, \mathrm{Dom})$ where $\mathcal{V}$ is the set of endogenous variables, the set $\mathcal{U}$ of exogenous variables, the set $\mathrm{F}$ of structural assignments (i.e., the functions mentioned above), and the set $\mathrm{Dom}$ of domains. In this formalism, $\forall X \in \mathcal{V} : X := \mathrm{F}_X(\mathrm{Pa}(X), U_X) \in \mathrm{Dom}(X)$, where $\mathrm{Pa}(X)$ denotes the inputs of $F_X \in \mathrm{F}$, which are called "causal parents" of $X$ and $U_X \in \mathcal{U}$ is the exogenous variable associated with $X$.*

In this paper, we focus on "recursive" SCMs, which assume a non-cyclic relationship of Pa, i.e., if $X$ is an ancestor of $Y$, then $Y$ cannot be an ancestor of $X$. It is therefore often represented via a Directed Acyclic Graph (DAG) called a causal graph. A context, denoted $u$, is a complete assignment of values to the exogenous variables, $u \in \mathrm{Dom}(\mathcal{U})$. Since the SCM is recursive, a context suffices to determine the values of all endogenous variables. To state that a certain variable $X \in \mathcal{V}$ took a certain value $x$ we denote: $(\mathcal{M}, u) \models (X = x)$. To model counterfactual reasoning, SCMs allow for interventions, i.e., forcing an

endogenous variable to take a value regardless of the output of its structural assignment. Interventions can lead to changes in the other variables. If forcing the value of $Y$ to $y'$ changed the value of $X$, we write: $(\mathcal{M}, u) \models [Y \leftarrow y'](X \neq x)$.

---

**Illustration**

*In example 1, the endogenous variables are ST (Suzy Throws), BT (Billy Throws), SH (Suzy Hits), BH (Billy Hits), and BS (Bottle Shatters). The exogenous variables are bt (Billy throw) and st (Suzy throw), which are set to 1 by assumption. The domains are boolean:* $\mathrm{Dom}(X) = \{0, 1\}$ *for any* $X \in \mathcal{V}$. *The following structural assignments model the system mechanisms:* $ST := st$, $BT := bt$, $SH := ST$, $BH := BT \wedge \neg SH$, *and* $BS := BH \vee SH$. *The observed values in context* $u = (1, 1)$ *are* $(ST, BT, SH, \neg BH, BS)$.

---

## 2.2 Actual causation and the HP-definition

Actual causation aims to identify which specific events in a particular situation were responsible for a given outcome. Notably, a cause is an event that took place before its consequence, such that if it did not occur, neither would the outcome. However, if Suzy had not thrown her rock, the bottle still would have shattered because of Billy's rock. To address this problematic case, the HP-definition (Halpern, 2015) includes a witness set $W$, composed of variables that are allowed to keep their actual values in the counterfactual world where the cause does not happen.

**Definition 2 (HP-cause)** *Let* $\mathcal{M} = (\mathcal{V}, \mathcal{U}, \mathrm{F}, \mathrm{Dom})$ *be an SCM and* $u \in \mathcal{U}$ *be a context for this SCM.* $C \subseteq \mathcal{V}$ *is an HP cause of* $T \in \{0, 1\}$ *in* $(\mathcal{M}, u)$ *if:*

*AC1* $(\mathcal{M}, u) \models (C = c^*)$ *and* $(\mathcal{M}, u) \models T$

*AC2* $\exists W \subseteq \mathcal{V} \backslash C, \exists c \neq c^*$ *s.t.* $(\mathcal{M}, u) \models (W = w^*)$ *and* $(\mathcal{M}, u) \models [C \leftarrow c, W \leftarrow w^*] \neg T$

*AC3 No subset of C satisfies the above conditions.*

Intuitively, AC1 ensures that the candidate cause and effect actually occurred, AC2 captures the idea that changing the cause (while holding certain variables fixed) would alter the effect, and AC3 guarantees that the cause is not unnecessarily large.

---

**Illustration**

*In our example,* $C = \{ST\}$ *is an HP cause of BS, with witness* $W = \{BH\}$ *since* $(\mathcal{M}, u) \models [ST \leftarrow 0, BH \leftarrow 0] \neg BS$. *The full set of causes is* $\mathcal{C} = \big\{\{ST\}, \{SH\}\big\}$.

---

## 2.3 Limitations and Challenges

The HP definition omits several considerations that are important for explanatory adequacy. It does not include notions such as normality (Halpern & Hitchcock, 2015) or degrees of responsibility (Chockler & Halpern, 2004), and it does not directly account for relevance (Reyd et al., 2025) or contrastiveness (Miller, 2021), which are central to human-oriented explanations. These limitations have motivated a large body of alternative definitions of actual causation. In this work, however, we focus on the identification problem and therefore adopt the standard HP definition, which is widely used as a solid formal basis. A more detailed discussion of these conceptual aspects is provided later in Section 7.

Identifying HP-causes poses significant intrinsic challenges. The HP definition requires reasoning over counterfactual worlds and subsets of variables, which leads to a combinatorial search space. Formally, deciding whether a candidate event is an actual cause is NP-complete (Halpern, 2015; Aleksandrowicz et al., 2017). This complexity arises directly from the structure of the definition—including the need to test alternative assignments (AC2) and to enforce minimality (AC3)—and affects even relatively small models.

These intrinsic difficulties motivate the need for practical methods to identify or approximate HP-causes. We next review prior work addressing actual-cause identification.

# 3    Related work

Existing approaches for the actual cause identification task can be broadly categorized into three families: exact methods using formal solvers, methods based on alternative causation definitions, and counterfactual explanation techniques from XAI. We review each in turn, highlighting their scope and limitations relative to the identification problem we address.

## 3.1    Exact HP cause identification

Early theoretical work on HP causation focused on conceptual definitions or complexity analysis rather than practical algorithms (Halpern, 2016). Hopkins (2002); Eiter & Lukasiewicz (2002); Aleksandrowicz et al. (2014); Halpern (2015); Aleksandrowicz et al. (2017) established the computational hardness of checking and identifying causes, but suggested only brute-force enumeration, feasible only for toy examples with few variables.

More recent efforts have leveraged constraint solvers to make brute-force search tractable. Ibrahim & Pretschner (2020) encoded the HP definition as SAT clauses, enabling automated verification of candidate causes using SAT solvers. Building on this, Ibrahim & Pretschner (2020) reformulated the problem as integer linear programming (ILP) to optimize for minimal causes, and developed tooling to automatically translate formal models of cyber systems into Boolean causal models (Ibrahim et al., 2020). While these solver-based methods represent significant progress, they remain restricted to Boolean domains and require complete knowledge of structural equations.

## 3.2    Alternative Definitions and Domain-Specific Methods

Researchers have proposed modified definitions, more adequate for a specific domain, alongside specialized identification algorithms. Albantakis et al. (2019) developed an information-theoretic notion of actual causation for discrete-time transition systems, while Rafieioskouei & Bonakdarpour (2024) adapted the HP framework to temporal logics with corresponding model-checking procedures. In reinforcement learning, Chuck et al. (2024) trained classifiers to identify functional causes, a relaxation of HP causes. Reyd et al. (2024) introduced a data-driven method that blends actual and general causation. These approaches successfully address identification in their target domains, but the definitional changes mean they do not solve the core HP identification problem.

## 3.3    Counterfactual explanations

The XAI literature on counterfactual explanations (Guidotti, 2024) addresses a closely related problem: given a model prediction, find alternative inputs that would yield different outputs. These methods typically minimize the distance between the original input and counterfactual instances. While conceptually similar to actual cause identification (Chou et al., 2022), counterfactual explanation methods differ in key ways. They focus on machine learning models rather than SCMs, seek single counterfactuals rather than exhaustive cause sets, and do not incorporate witness sets $W$ from the HP definition. Specialized extensions exist: backtracking counterfactuals (Kladny et al., 2024) reason about exogenous variables in SCMs, while Karimi et al. (2020) adapts algorithmic recourse (counterfactual explanations focusing on the actionability of the counterfactual) to SCMs for specific model classes. However, none of these methods is easily adaptable to full HP identification across SCM types.

Overall, the literature reveals a clear gap: no existing method practically identifies the complete set of HP causes, notably for non-Boolean and stochastic SCMs. Solver-based methods are restricted to Boolean domains with known equations. Specialized definitions are limited to specific use-cases. Counterfactual explanation techniques lack the HP definition's witness sets and can't target larger causes. We now introduce algorithms designed to fill this gap.

# 4    Our algorithms

We present three complementary algorithms for identifying HP-causes with polynomial-time complexity and adjustable precision. The Minimal Beam Search (MBS) algorithm serves as our baseline, requiring only an oracle to evaluate counterfactual outcomes and a heuristic to guide the search. The Iterative Sub-Instance (ISI) algorithm improves upon MBS by exploiting causal graph structure, when available, to decompose the search space into smaller subproblems; an approach justified by our theoretical result (Theorem 1) showing that this decomposition preserves the set of HP-causes. The Lower-Upper Confidence Bound (LUCB) algorithm addresses stochastic models by adaptively allocating samples to interventions where the cancellation status or heuristic value remains uncertain, substantially reducing the number of model evaluations needed. While each algorithm can be used independently, they combine naturally: ISI can call MBS to solve subproblems, and LUCB can replace deterministic evaluation in either MBS or ISI when the model is stochastic.

## 4.1    Minimal Beam Search (MBS)

### 4.1.1    Problem Formulation and Assumptions

MBS relaxes the restrictive assumptions of prior HP-cause identification methods. Unlike solver-based approaches (Ibrahim & Pretschner, 2020), which require complete knowledge of structural assignments and restrict to Boolean domains, MBS requires only: (1) the set of endogenous variables $\mathcal{V}$, (2) their domains Dom, (3) an oracle function $\phi$ which evaluates whether a given intervention $e$ cancels the target consequence, i.e., $\phi(e) = 1$ if $(\mathcal{M}, u) \models [e](T)$ and $\phi(e) = 0$ if $(\mathcal{M}, u) \models [e](\neg T)$, and (4) a heuristic function $\psi$ that estimates how "close" an intervention is to canceling $T$, guiding the search toward promising regions.

The oracle can be realized in multiple ways: by applying structural assignments when available, by querying a simulator that supports interventions, or by experimenting on a controllable physical system. The heuristic embeds domain knowledge about what makes $T$ likely or unlikely.

We acknowledge that the heuristic is a significant practical requirement. Unlike the oracle, $\psi$ must be crafted for each problem domain. We provide empirical comparisons of some boolean heuristic choices in Annex E but do not offer general construction guidelines. Additionally, Section 7.1 presents an example of a domain-guided heuristic that illustrates what a "promising" intervention is, i.e., how to evaluate if an intervention is close to canceling $T$.

This remains a limitation of our approach and an avenue for future work.

Let $e$ denote an intervention, formalized as a set of variable-value pairs: $e = \{(Y_i, y_i)\}_{i=0}^{n}$ where $Y_i \in V$, $y_i \in \text{Dom}(Y_i)$, and $Y_i$ are distinct. We partition the variables in $e$ into the counterfactual variables $Ctf(e)$ and the actual variables $Act(e)$. For Boolean models, we use compact notation: $\{\mathbf{Y}\}$ for $\{(Y, 1)\}$ and $\{\neg\mathbf{Y}\}$ for $\{(Y, 0)\}$; we use bold notation to differentiate the positive value from the variable. For any HP-cause $C$ with witness $W$, there exists an intervention $e$ such that $Ctf(e) = C$ and $Act(e) = W$.

> **Illustration**
>
> *In the rock-throwing scenario (Example 1), consider $e_1 = \{\neg\mathbf{BT}\}$ and $e_2 = \{\neg\mathbf{SH}, \neg\mathbf{BH}\}$. We have $Ctf(e_1) = \{BT\}$ and $\phi(e_1) = 1$ (bottle still shatters), so $e_1$ does not identify a cause. For $e_2$, $Ctf(e_2) = \{SH\}$, $Act(e_2) = \{BH\}$, and $\phi(e_2) = 0$, identifying $\{SH\}$ as a cause with witness $\{BH\}$.*

### 4.1.2    The MBS algorithm

MBS explores the space of interventions to identify all pairs (C, W) satisfying AC2 (counterfactual dependence) and AC3 (minimality). The algorithm adapts classical beam search (Lowerre, 1976). At each step $k$, Beam Search maintains a beam of $b$ interventions, i.e., those with the best heuristic values $\psi(e)$. Each intervention $e$ in the beam is expanded by adding one variable-value pair (counterfactual or actual) for each variable not in $e$. The algorithm terminates when all variables have been considered or when an early-stop condition is met (at least one cause found and no promising interventions remain).

We adapt Beam Search to HP-cause identification using three "minimality improvements". MBS initializes only counterfactual values: it prevents evaluating interventions that have no effect. MBS prunes non-canceling branches: once $\phi(e) = 0$, we do not expand $e$ further, as any extension would be a superset of an already-identified cause. MBS enforces minimality during expansion: when generating candidate interventions for step $k + 1$, it discards any $e$ where $Ctf(e)$ is a superset of an already-identified cause.

Algorithm 1 provides the full pseudo-code; we summarize key steps here. The algorithm iterates for at most `max_steps` iterations, where `max_steps` is a settable parameter. At each iteration, the beam is expanded (line 4), producing new interventions that respect minimality constraints. Interventions are evaluated with $\phi$ (line 7), and those with $\phi(e) = 0$ are filtered for minimality and added to the output set $C$ (line 8). Minimality filtering happened once during the beam expansion to avoid evaluating interventions that would not be minimal with respect to previously identified causes. Another filtering is necessary (here line 8) to ensure that all causes identified during the current step are indeed minimal with respect to each other (see step 2 of the example in Appendix A). The remaining interventions ($\phi(e) = 1$) are scored by $\psi$, and the top b are retained for the next iteration (line 9).

---

**Algorithm 1:** MBS Algorithm

**Input** : $\mathcal{V}$, $v^*$, Dom, $\phi$ (includes the target $T$), $\psi$, beam size $b$, early_stop, max_steps
**Output:** List of all causes

1 $C \leftarrow \emptyset$ ;                                                    // Identified causes
2 $B \leftarrow \emptyset$ ;                                                    // Beam made of interventions $e$
3 **for** $t \leftarrow 1$ **to** max_steps **do**
4     $B \leftarrow$ expandBeam$(B, V, D, v^*, C)$;        // Interventions $e$ with $Ctf(e)$ minimal with $C$
5     **if** $B = \emptyset$ *or* (early_stop & $C \neq \emptyset$) **do**
6        break;
7     neg, pos $\leftarrow$ splitEval$(\phi(B))$;                // Distinguished $\phi(e) = 1$ from $\phi(e) = 0$
8     $C \leftarrow C \cup$ filterMinimality$(\text{neg}, C, v^*)$;
9     $B \leftarrow$ getBest$(\psi(\text{pos}), b, C)$;            // Keep $b$ best for $\psi$ with $Ctf(e)$ minimal in $C$
10 **return** $C$;

---

**Illustration**

*We start with interventions $\{\neg\textbf{BT}\}$, $\{\neg\textbf{ST}\}$, $\{\textbf{BH}\}$, and $\{\neg\textbf{SH}\}$ that all, have $\phi(e) = 1$. Using heuristic $\psi = $ "minimize positive variables" and beam size $b = 3$, we retain $\{\neg\textbf{BT}\}$, $\{\neg\textbf{ST}\}$, and $\{\neg\textbf{SH}\}$. Among the 18 expanded candidate, $\{\neg\textbf{ST}, \neg\textbf{BT}\}$ and $\{\neg\textbf{ST}, \neg\textbf{BH}\}$ have $\phi(e) = 0$. Since $Ctf(\{\neg\textbf{ST}, \neg\textbf{BT}\}) = \{ST\} \subseteq Ctf(\{\neg\textbf{ST}, \neg\textbf{BH}\}) = \{ST, BH\}$, only $C = \{ST\}$ qualifies as a cause with witness $W = \{BH\}$. See Annex A for the full execution trace.*

## 4.2 Iterative Sub-Instance decomposition (ISI)

The ISI algorithm leverages knowledge of the causal graph (DAG) to divide the search space into smaller, more tractable subproblems. This introduces an additional assumption beyond MBS—access to the DAG structure, but substantially improves efficiency when available.

### 4.2.1 Intuition and motivating examples

The key insight is that for any cause $C$, the causal effect must propagate from $C$ to $T$ through intermediate variables (descendants of $C$). Rather than searching for causes among all variables simultaneously, we can first search for causes in $\text{Pa}(T)$, and then, for each identified cause, perform a "causal backtrack," i.e., search in its parents. Finally, we can recursively repeat until no parents remain.

If one wanted to perform this "causal backtracking" naively, one could only search within the parent of the identified cause. However, this naive "causal backtracking" fails in several cases presented in the five

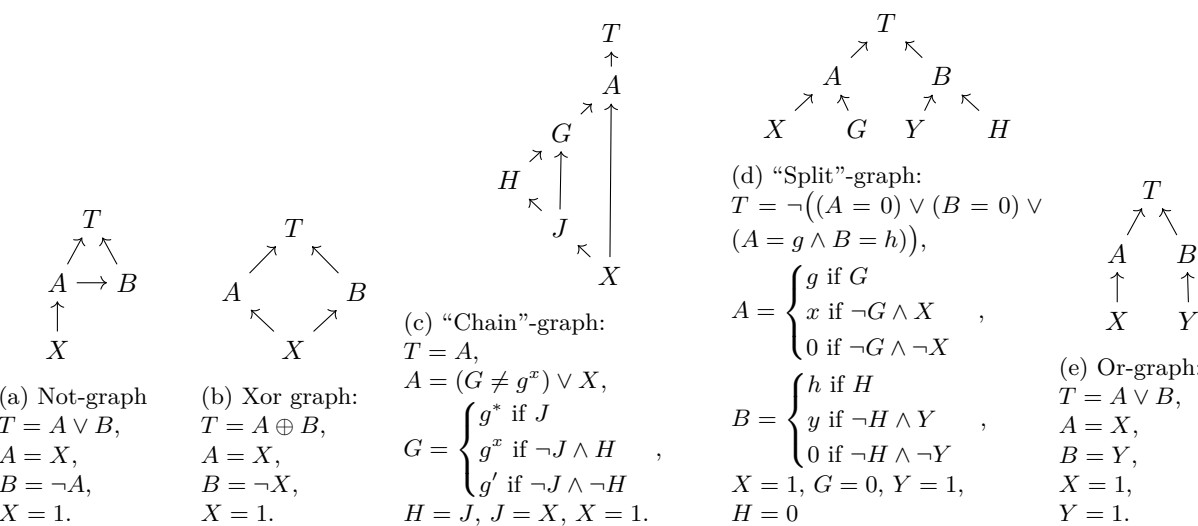

Figure 1: Five SCMs to understand how ISI "backtracks" from an identified cause.

scenarios in Figure 1. In a "NOT"-structure such as Figure 1a, if $A$ is a cause with witness $B$, then "causally backtracking" to search in $\mathrm{Pa}(A)$ must preserve $B$ as a witness; otherwise $X$ does not cancel $T$ (in addition to searching within the parent of the previously identified cause, we must retain the current witness). In a "XOR"-structure such as Figure 1b, if $A$ is identified as a cause, "causal backtracking" requires blocking the alternate path through $B$ (by fixing $B$ to its actual value) to reveal $X$ as the underlying cause (in addition to searching within the parent of the previously identified cause, we must add the alternate path to the target as witnesses). In a "chain"-structure such as Figure 1c, "causal backtracking" requires considering some intermediate variables as witnesses, as variables between $X$ and $A$ may block or open causal paths depending on their values; here $H$ is a witness for $X$, contrary to $J$ (in addition to searching within the parent of the previously identified cause, we must add their descendant that are ancestor of the target as optional witnesses). In a "split"-structure such as Figure 1d, "causally backtracking" from $A$ or $B$ individually fails, and we must "causally backtrack" from the non-minimal "canceling" intervention $\{A, B\}$ to identify $\{G, H\}$ (in addition to searching within the parents of the previously identified cause, we must search within the parents of the non-minimal "canceling" intervention). Finally, in an "OR"-structure such as Figure 1e, "causal backtracking" from $\{A, B\}$ requires considering all its subsets independently to identify causes such as $\{X, B\}$ (in addition to searching within the parent of the previously identified cause, we must search within all their non empty subsets). These examples motivate the formal construction that follows.

### 4.2.2 Subproblem Construction

Given a "canceling" intervention $(C, c)$ with witness $W$, we define a subproblem for each non-empty subset $S \subseteq C$ by specifying:

- $I = \mathrm{Pa}(S) \setminus C$: the free search space (variables we'll search over)
- $W_0 = \mathrm{De}^-(I) \cap \mathrm{An}^-(C)$: optional witness (descendants of I that are ancestors of C)
- $R_C = C \setminus S$: forced counterfactual interventions (parts of C not being "backtracked")
- $r_C = c_{|R_C}$: values for the forced interventions
- $R_W = W \cup \mathrm{De}^{-\setminus C}(I)$: forced witness (original witness plus alternate paths from $I$ to $T$)

where $\mathrm{De}^-(Y)$ denotes strict descendants $(\mathrm{De}(Y) \setminus Y)$ and $\mathrm{De}^{-\setminus C}(I)$ denotes $\mathrm{De}^-(I) \setminus \big(\mathrm{De}(C) \cup C \cup \mathrm{An}(C)\big)$. A subproblem $\kappa = (I, W_0, R_C, r_C, R_W)$ constrains the search: we seek interventions on variables in $I$ (with actual values optionally on $W_0$) that, combined with forced interventions $R_C \leftarrow r_C$ and $R_W \leftarrow r_W^*$, cancel $T$. The forced witness $R_W$ is associated with its actual values $r_W^*$ that are given by the context $(\mathcal{M}, u) \models (R_W = r_W^*)$ and don't need to be specified in the subproblem. The examples in Figure 1 give the intuition

---

**Algorithm 2:** ISI algorithm

---

**Input** : $\mathcal{V}$, DAG over $\mathcal{V}$, `search` an algorithm that finds "canceling" counterfactual interventions (with a witness)

**Output:** List of all causes

**1** $\mathbf{C} \leftarrow \emptyset$;

**2** $Q \leftarrow \emptyset$;

**3** $Q$.queue$((\text{Pa}(T), \emptyset, \emptyset, \emptyset, \emptyset))$;

**4** **while** $Q \neq \emptyset$ **do**

**5**     $(I, W_0, R_C, r_C, R_W) \leftarrow Q$.dequeue();

**6**     **for each** $(C, c, W) \in \text{search}(I, W_0, R_C, r_C, R_W)$ **do**

**7**        $\mathbf{C} \leftarrow \text{minimalUpdate}(\mathbf{C}, C)$;

**8**        **for each** $S \in 2^C \setminus \emptyset$ **do**

**9**           $I' \leftarrow \text{Pa}(S) \setminus C$;

**10**           $W_0' \leftarrow \text{De}^-(I') \cap \text{An}^-(C)$;

**11**           $R_C' \leftarrow C \setminus S$;

**12**           $r_C' \leftarrow c_{|R_C'}$;

**13**           $R_W' \leftarrow W \cup \big(\text{De}^-(I') \setminus (\text{De}(C) \cup C \cup \text{An}(C))\big)$;

**14**           $Q$.add$((I', W_0', R_C', r_C', R_W'))$;

**15** **return** $\mathbf{C}$;

---

behind each part of these formulas. Instead of naively backtracking and simply searching within the parents of the cause, we select a subset of it, replace it with its parents (as prompted by Figure 1e). We then consider as an optional witness their descendant that are ancestors of the cause (as prompted by Figure 1c). We then consider as forced intervention the remainder of the cause (still as prompted by Figure 1e). Finally, we force as witness the original witness (as prompted by Figure 1a) and the alternate path from $I$ to $T$ (as prompted by Figure 1b). These example provide intuitions and illustrations; the proofs of why these exact formulas work are presented in Appendix F.

### 4.2.3 The ISI algorithm

We initialize the algorithm with the subproblem $(\text{Pa}(T), \emptyset, \emptyset, \emptyset, \emptyset)$, i.e., only variables in $\text{Pa}(T)$ are considered as part of the search space, using their full domain, and no additional constrained interventions are added. We also initialize an empty set of causes $\mathbf{C}$. We then use a queue to implement the recursive construction of "subproblems". As long as the queue contains elements, we dequeue a new "subproblem" described by $(I, W_0, R_C, r_C, R_W)$. We then identify the set of "canceling" counterfactual interventions using some provided search procedure. For each "canceling" counterfactual intervention $C \leftarrow c$, if $C$ is minimal in $\mathbf{C}$, we add it to $\mathbf{C}$ and remove it supersets. We then iterate over non-empty subsets $S \subseteq C$. For each $S$, we build a new "subproblem" described by $I' = \text{Pa}(S) \setminus C$, $W_0' = \text{De}^-(I') \cap \text{An}^-(C)$, $R_C' = C \setminus S$, $r_C' = c_{|R_C'}$, and $R_W' = W \cup \text{De}^{-\setminus C}(I')$. We finally, add $(I', W_0', R_C', r_C', R_W')$ to the queue.

When the SCM is Boolean, ISI can "causally backtrack" only from minimal "canceling" interventions (i.e., HP-causes identified in the current subproblem) rather than all "canceling" interventions. This substantially reduces the number of subproblems generated and accelerates convergence.

> **Illustration**
>
> On Example 1, ISI begins with $I = \{SH, BH\}$ and identifies *the "canceling" intervention* $\{\mathbf{SH}, \neg\mathbf{BH}\}$, *with cause* $C_1 = \{SH\}$ *and witness* $W = \{BH\}$. "Causally backtracking" with $S = \{SH\}$ yields $I' = \{ST\}$, $R_W = \{BH\}$. Searching in $\{ST\}$ with enforced $BH \leftarrow 0$ identifies $C_2 = \{ST\}$ as a cause. Since $ST$ has no parents, the algorithm terminates, having found both $C_1 = \{SH\}$ and $C_2 = \{ST\}$.

### 4.2.4 Correctness guarantees and practical use

The following two theorems justify the ISI construction and show that it exhaustively recovers all HP-causes. We present them informally here. The formalized versions and full proofs can be found in Annex F.

**Theorem 1 (ISI Completeness)** *For any SCM, every HP-cause C appears as the variable set of some "canceling" intervention generated by the exact ISI algorithm.*

**Theorem 2 (Boolean Completeness)** *When the SCM is Boolean, "causally backtracking" only from minimal "canceling" interventions (HP-causes) remains complete: every HP-cause is still discovered.*

These guarantees apply when ISI performs an exhaustive search in each subproblem. In practice, using MBS with a finite beam size $b > 0$ yields an approximate ISI that trades completeness for efficiency. ISI greatly reduces the search space, which results in significant potential performance gain (both in terms of quality and efficiency). However, ISI requires additional assumptions. We require the causal graph, which was not necessary for MBS. When the system is not Boolean, we remove the minimality improvements of MBS, which can increase the runtime. For each "canceling" intervention, we consider all its subsets, which can be intractable for large interventions. Section 6 discusses the impact of using ISI over MBS, and Section 7 suggests additional approximations for the ISI algorithm to address these considerations.

## 4.3 Adaptive Sampling for Stochastic Models (LUCB)

When the SCM is stochastic, either inherently or because interventions are modeled probabilistically (e.g., aggregated variables set via random micro-level actions), the oracle $\phi$ and heuristic $\psi$ return random values. A naive approach would consist of sampling each intervention a fixed N times and applying MBS or ISI to the estimates $\widehat{\phi}(e)$ and $\widehat{\psi}(e)$, where an intervention is considered to "cancel" $T$ when $\phi$ is below a certain "canceling" threshold $a$. This would be wasteful: most interventions can be confidently classified with far fewer samples, while borderline cases require extensive sampling. LUCB addresses this by maintaining statistical confidence intervals (upper and lower bounds) around each estimate and adaptively resampling only interventions whose bounds remain ambiguous relative to decision thresholds, accounting for small tolerance slacks $\epsilon$ that prevent excessive sampling on near-ties.

The LUCB algorithm adapts the Lower-Upper Confidence Bound framework (Kaufmann & Kalyanakrishnan, 2013) to allocate samples adaptively, focusing computational effort on interventions whose status remains uncertain. This approach was successfully employed in the Anchors explanation method (Ribeiro et al., 2018) for a similar beam-selection problem.

### 4.3.1 Confidence Bounds and decision criteria

For each intervention $e$, LUCB maintains empirical estimates $\widehat{\phi}(e)$ and $\widehat{\psi}(e)$ along with confidence intervals constructed from concentration inequalities. Because $\phi(e)$ is Bernoulli (binary outcome) and $\psi(e)$ is real-valued with unknown variance, we employ different bounds.

For $\phi$, we use the Chernoff concentration (Boucheron et al., 2013) to compute the tightest intervals $[L_\phi(e), U_\phi(e)]$ such that $P\big(L_\phi(e) \leq E[\phi(e)] \leq U_\phi(e)\big) \geq 1 - \delta$. With $n(e)$ the number of samples for $\psi(e)$, The upper bound $U_\phi(e)$ is then the largest $q \geq \widehat{\phi}(e)$ satisfying

$$n(e)\mathrm{KL}(\widehat{\phi}(e)\|q) = \log \frac{2}{\delta},$$

and $L_\phi(e)$ is the smallest $q \leq \widehat{\phi}(e)$ solving the same equation (Garivier & Cappé, 2011). In practice, these bounds are computed via a simple monotone numerical root-finding procedure on $q \in [0, 1]$. An arm is confidently "cancelling" $T$ if $U_\phi(e) < a + \epsilon_c$ and confidently "non-cancelling" if $L_\phi(e) > a - \epsilon_{nc}$, where $a$ is the "canceling" threshold. Arms not satisfying either inequality remain ambiguous.

For $\psi$, we use the Bernstein inequality to derive the empirical Bernstein bounds (Maurer & Pontil, 2009). With $s^2(e)$ the empirical variance of $\psi$, $\delta$ the confidence level and $\beta = \log 2/\delta$, we have:

$$U_\psi(e) = \widehat{\psi}(e) + \sqrt{\frac{2s^2(e)\,\beta}{n(e)}} + \frac{3\,\beta}{n(e)}, \qquad L_\psi(e) = \widehat{\psi}(e) - \sqrt{\frac{2s^2(e)\,\beta}{n(e)}} - \frac{3\,\beta}{n(e)}.$$

Let $B$ be the beam composed of the $b$ (beam-size) interventions with the smallest empirical $\widehat{\psi}$ and satisfying $\widehat{\phi}(e) \geq a$. We identify the highest upper bound in the beam $u = \arg\max_{e \in B} U_\psi(e)$ and the lowest lower bound outside the beam $\ell = \arg\min_{e \notin B} L_\psi(e)$. The beam is certified when $U_\psi(u) - L_\psi(\ell) \leq \epsilon_b$. If this condition does not hold, we identify the set of disputed arms which are resampled in the next batch. Given a set of interventions $E$ to evaluate, the set of unsure ones can be written: $S = \{e \in E : L_\psi(e) \leq U_\psi(u) + \varepsilon_b \text{ and } U_\psi(e) \geq L_\psi(\ell) - \varepsilon_b\}$.

LUCB addresses two distinct decisions for each intervention $e$: whether it confidently cancels $T$ (i.e., whether $\mathbb{E}[\phi(e)]$ falls below the threshold $a$), and whether it belongs in the beam (i.e., whether it ranks among the $b$ interventions with the smallest $\psi$). Neither is a ranking problem — both are thresholding and membership decisions — so the goal is not to find the single best intervention but to certify, with high probability, which interventions fall on which side of each boundary. The confidence-bound framework is natural here precisely because it enables *adaptive* sampling: interventions whose status is already clear require no further samples, while only the ambiguous borderline cases are resampled.

### 4.3.2 Execution Flow.

After an initial sampling phase in which each intervention is evaluated a fixed number of times, the algorithm proceeds iteratively by alternating between: (i) identifying unsure arms that require resampling and running the new batches, (ii) updating the bounds using the newly acquired statistics, and (iii) updating borderline indices and computing the stopping criterion. The loop terminates once both the cancelling decisions and the beam membership are stable within their respective tolerances or when a predefined computational budget is reached. Algorithm 3 illustrates this process.

---

**Algorithm 3:** LUCB algorithm

---

**Input**  : Beam size $b$, A set of interventions $E$, Cancellation threshold $a$, Slacks $\epsilon_b$, $\epsilon_c$, $\epsilon_{nc}$, $\phi$, $\psi$,
           `init_batch_size`, `batch_size`, average budget per arm $N$

**Output:** List of all causes

1   $\widehat{\phi}(E), \widehat{\psi}(E), s^2(E), n(E) \leftarrow$ `sample(`$E$`, init_batch_size, `$\phi$`, `$\psi$`)`;

2   $L_\phi(E), L_\psi(E) \leftarrow 0$;

3   $U_\phi(E), U_\psi(E) \leftarrow 1$;

4   $c, nc, u, \ell \leftarrow$ `findBorderline(`$L_\phi(E), U_\phi(E), L_\psi(E), U_\psi(E)$`)`;

5   **while** $U_\phi(c) < a + \epsilon_{nc}$ *or* $L_\phi(nc) > a - \epsilon_c$ *or* $U_\psi(u) - L_\psi(\ell) \leq \epsilon_b$ **do**

6      $S \leftarrow$ `findUnsure(`$L_\phi(E), U_\phi(E), L_\psi(E), U_\psi(E)$`)`;

7      $\widehat{\phi}(S), \widehat{\psi}(S), s^2(S), n(S) \leftarrow$ `sample(`$S$`, batch_size, `$\phi$`, `$\psi$`)`;

8      $L_\phi(S), U_\phi(S), L_\psi(S), U_\psi(S) \leftarrow$ `computeBounds(`$\widehat{\phi}(S), \widehat{\psi}(S), s^2(S), n(S)$`)`;

9      $c, nc, u, \ell \leftarrow$ `findBorderline(`$L_\phi(E), U_\phi(E), L_\psi(E), U_\psi(E)$`)`;

10     **if** $\sum_{e \in E} n(e) > N \times |E|$ **do**

11       |   break;

12 **return** $\widehat{\phi}(E), \widehat{\psi}(E)$;

---

## 5 Experiments

This section presents the SCMs used to evaluate the proposed algorithms and describes the experimental protocol and evaluation metrics. The experiments are designed to assess three aspects of our contributions: (i) the ability of the proposed baseline to approximate HP-cause identification, (ii) the impact of complemen-

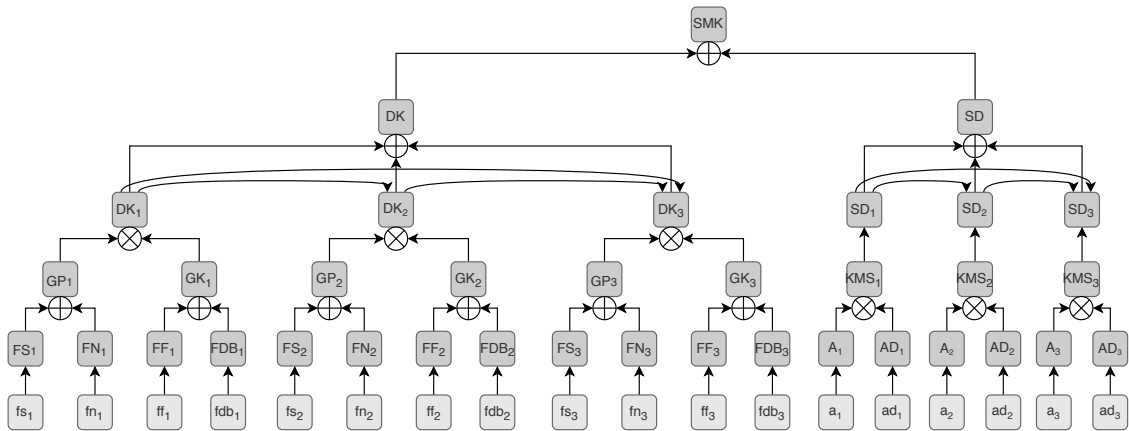

Figure 2: Diagram for the original SMK scenario with 3 attackers.

tary algorithms on efficiency and scalability, and (iii) the applicability of our approach beyond deterministic Boolean models.

Because no standard benchmarks exist for HP-cause identification beyond highly restricted settings, we focus on a single family of SCMs that can be systematically varied in size, determinism, and variable domains. This controlled setup allows us to isolate the behavior of the algorithms under comparable conditions. In addition, we report sanity checks on small SCMs from the philosophical actual causation literature in Annex B, which constitutes the only available way to validate correctness on well-established conceptual examples.

All SCMs used in the main experiments derive from a base scenario introduced by Ibrahim & Pretschner (2020). We first present the base SCM modeling this scenario and then describe two modified versions, a non-Boolean version and a stochastic version, introduced to illustrate the wider range of applicability of our methods.

### 5.1 Models

**Base SCM.** The SMK scenario supposes that $k$ attackers attempt to obtain the master key of a computer. The target predicate $SMK$ is true if an attacker steals or decrypts the master key. These events are observed via the Boolean variables $SD$ (steal decrypted) and $DK$ (decrypt key), which are true if at least one attacker performs the corresponding action. For each attacker $i$, child variables model the different ways in which the key or the passphrase can be obtained. An attacker $i$ can steal or decrypt the key only if no previous attacker has already done so. For instance, if attackers 1 and 2 both meet the conditions to steal the key, then $SD_1$ is true, and $SD_2$ is false. The key is decrypted if an attacker $i$ gets the key ($GK_i$) and the passphrase ($GP_i$). The key is stolen if an attacker $i$ gets it from Key Management Service ($KMS_i$). An attacker $i$ can obtain the key from a file ($FF_i$) or a database ($FDB_i$), and can get the password from the script ($FS_i$) or the network ($FN_i$). To obtain the key from $KMS$, an attacker $i$ must have access to it ($A_i$) and attach a debugger ($AD_i$). The structural equations for $n$ attackers are as follows: : $\mathbf{SMK} = DK \lor SD$, $\mathbf{DK} = DK_1 \lor ... \lor DK_n$, $\mathbf{SD} = SD_1 \lor ... \lor SD_n$, $\forall i : \mathbf{DK_i} = GP_i \land GK_i \land \bigwedge_{j \in [\![1,i-1]\!]} \neg DK_j$, $\mathbf{SD_i} = KMS_i \land \bigwedge_{j \in [\![1,i-1]\!]} \neg SD_j$, $\mathbf{GP_i} = FS_i \lor FN_i$, $\mathbf{GK_i} = FF_i \lor FDB_i$, and $\mathbf{KMS_i} = A_i \land AD_i$. The DAG with visual insights into the structural functions is shown in Figure 2.

**Non-boolean SCM.** We introduce a modified version of the SMK SCM that models the same scenario with a different representation. In this new version, variables $GP_i$ and $GK_i$ take values in larger discrete domains and are computed using summation rather than Boolean operators: $\mathrm{Dom}(GP_i) = \mathrm{Dom}(GK_i) = \{0, 1, 2\}$. Variables $DK_i$ retain their Boolean domains and are defined as $\mathbf{DK_i} = GP_i > 0 \land GK_i > 0 \land \bigwedge_{j \in [\![1,i-1]\!]} \neg DK_j$.

**Stochastic SCM.** To evaluate our algorithms in a stochastic setting, we further modify the SMK SCM by introducing noise into the structural equations. Each variable has a probability $\epsilon_n$ (the noise level) of flipping the value prescribed by its structural function. Concretely, each variable follows a Bernoulli distribution $\mathcal{B}(\epsilon_n)$ or $\mathcal{B}(1 - \epsilon_n)$:

$$X \sim \begin{cases} \mathcal{B}(1 - \epsilon_n) & \text{if } F_X(PA_X, U_X) = 1 \\ \mathcal{B}(\epsilon_n) & \text{otherwise} \end{cases}$$

## 5.2 Experimental setup

We test our algorithms on several sizes of the SMK model. For each experiment, we consider models with 2, 5, and 10 attackers. For each configuration, we generate 20 unique contexts by repeatedly sampling exogenous variables and retaining only contexts in which half of the exogenous variables are true, and half are false, and for which the outcome $SMK$ is true. This ensures comparable and balanced contexts across runs.

To evaluate full HP-cause identification, we run the algorithms with beam sizes growing exponentially from 2 to 256, across all model variants (Boolean, non-Boolean, and stochastic), for each context and algorithm (MBS and ISI). We set `max_steps` to 7, which is another source of approximation our algorithm allows. For stochastic SCMs, each experiment is repeated with 10 different random seeds, and results are averaged across seeds. We use slacks of 10%, a canceling threshold $a = 0.65$, a confidence level $\delta = 10\%$, $N = 50$ samples per intervention, an initial batch size of 30, and a batch size of 10. The noise level is set to $\epsilon_n = 1.5/|V|$, yielding an average of 1.5 value flips per model evaluation.

For the smallest cause identification task, we run both of our algorithms with early stopping, with no maximum number of steps, using the same beam sizes and considering models with 2 to 15 attackers. We also compare our results to the approach of Ibrahim & Pretschner (2020), which we reimplemented using their ILP formulation and the Gurobi solver. Because we rely on the free version of Gurobi, we are limited to at most 7 attackers.

Across all experiments, we use the same base heuristic, which minimizes the number of variables with an observed value of 1. In the non-Boolean setting, Boolean variables are treated as integers, and the heuristic sums the values of all exogenous variables. Additional results for alternative heuristics are reported in Annex E.

## 5.3 Evaluation

To evaluate the smallest-cause identification task, we compute the expected size of the smallest cause as the sum of $SD$ and $DK$ in the actual state. We report the accuracy score as 1 when an algorithm identifies a smallest cause of the correct size and 0 otherwise.

For the full identification task, the algorithms output a set of causes. We evaluate it by comparing this set to a reference set of expected causes. The reference set is obtained by running the ISI algorithm with beam size $b = -1$ and no maximum number of steps, which corresponds to exact HP-cause identification. The stochastic and non-Boolean versions share the same solutions as their deterministic Boolean counterpart. Hence, we used the same set of contexts for the different SCM versions and therefore reused the same reference sets. On the Boolean SCMs considered in our experiments (with 2, 5, and 10 attackers), this exact identification is tractable and can be computed in practice using the ISI algorithm. Set similarity is measured using the DICE score: $dice(A, B) = \frac{2|A \cap B|}{|A| + |B|}$. We report DICE scores in % for better readability.

In addition to quality metrics, we report the number of calls to the model, which reflects the computational cost of our approach while remaining independent of the internal implementation or complexity of the model. In our experiments, one model call corresponds to a single evaluation of both $\phi$ and $\psi$. For comparisons with the ILP-based approach, we report runtime instead, as that method does not rely on explicit model calls.

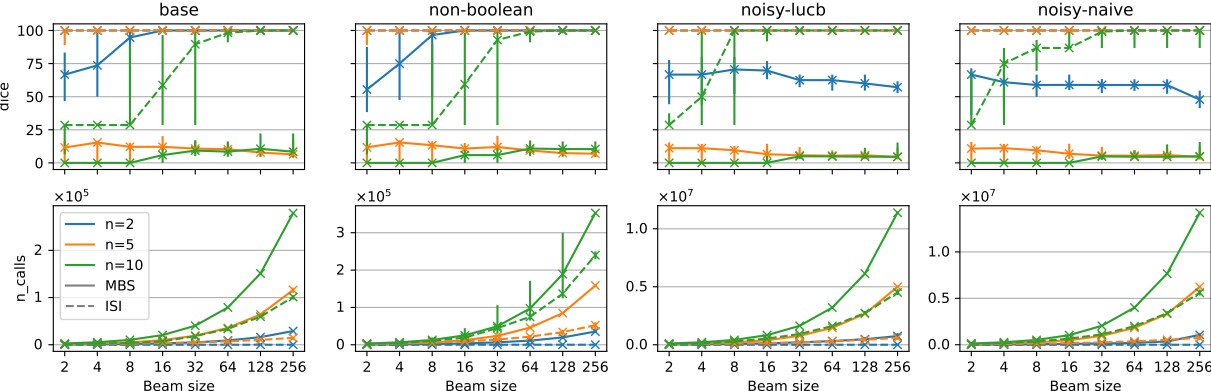

Figure 3: DICE score and number of calls for full causes identification in our three versions of SMK (column 1: base Boolean model; column 2: non-Boolean model; columns 3 and 4: stochastic model). The line color corresponds to the number of attackers. The linestyle corresponds to the identification algorithm. The x-axis uses a logarithmic scale. The bottom plots do not share the same scale. The lines report the median values for the 20 contexts and the error bars report the 25-75 quantiles.

We summarize the 20 contexts by their median and report the 25–75 quantiles (interquartile range) as error bars, both for the quality measures (DICE) and for the number of calls. The only exception is the binary accuracy metric, which we summarize by its mean. We use the interquartile range rather than the standard deviation because the variance can be large — some contexts are more challenging than others, and small changes in the identified cause set (e.g., missing a single cause) can substantially affect the DICE score by generating many supersets — so standard-deviation bars would be hard to read and could even exceed the natural [0, 100] range of the DICE score.

## 6 Results

Our experiments support four main findings: (1) the beam size parameter provides explicit control over the trade-off between quality and computational cost; (2) ISI substantially improves full HP-cause identification compared to MBS; (3) ISI identifies the smallest cause more efficiently than both ILP and MBS; (4) LUCB reduces the number of model calls in stochastic settings with limited impact on quality.

### 6.1 The beam size parameter

Figure 3 shows that across all settings, increasing the beam size monotonically increases or preserves both quality and cost. The number of model calls follows a clear exponential trend, which is consistent with a quadratic complexity when accounting for the logarithmic scale of the x-axis (see Annex D for details). The DICE score increases clearly in several configurations, notably for MBS with $n = 2$ and for ISI with $n = 10$ in the deterministic models. In other cases, the DICE score remains flat, particularly for MBS in the noisy model with $n = 2$ and in deterministic models with $n \leq 5$. This suggests that the baseline algorithm struggles in these settings and would require larger beam sizes to improve.

Quantitatively, doubling the beam size increases the DICE score by up to 19.1 points for MBS (from $b = 8$ to $b = 16$ in the base model with $n = 2$) and up to 13.5 points for ISI (from $b = 8$ to $b = 16$ in the base model with $n = 10$).

Figure 4 illustrates three representative cases where increasing the beam size improves quality. We plot the quality measure (accuracy in Figure 4c and DICE in Figures 4a and 4b) against runtime. These plots illustrate how the beam size acts as a practical lever to control the quality–efficiency trade-off.

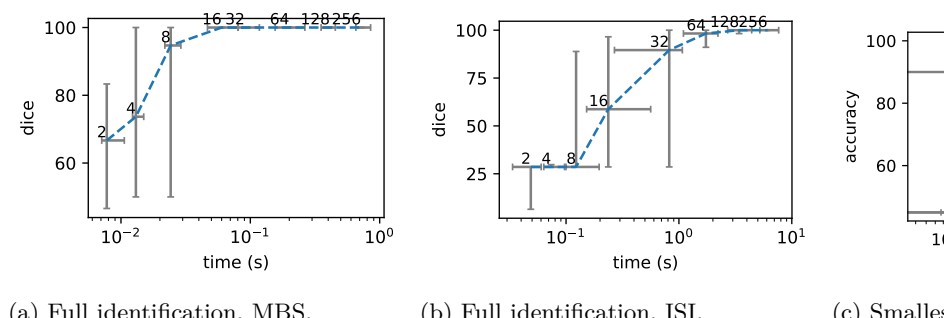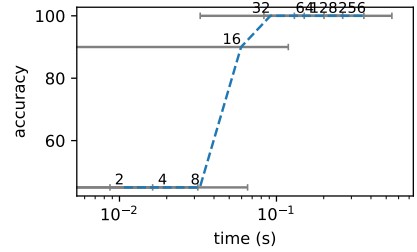

(a) Full identification, MBS, base model, $n = 2$.

(b) Full identification, ISI, base model, $n = 10$.

(c) Smallest identification, MBS, base model, $n = 10$.

Figure 4: Tradeoff between quality of the identified causes and number of calls to the model. The x-axis uses a logarithmic scale. The bottom plots do not share the same scale. The lines report the median for 20 contexts for the left and center figures and the mean for the right figure (as accuracy is a binary metric). Error bars report the 25-75 quantiles for the first two figures and standard deviation for the third. There is no error bar on the y-axis of the third figure as accuracy is binary.

## 6.2 Impact of the ISI algorithm on full identification

For $n \leq 5$, Figure 3 shows that ISI achieves near-perfect DICE scores across all beam sizes. For $n = 10$, the DICE score starts lower but increases steadily as the beam size grows.

In the Boolean model, ISI "causally backtracks" directly on the smallest canceling counterfactual interventions, which correspond to HP causes. As a result, ISI is extremely efficient and requires orders of magnitude fewer model calls than MBS. Notably, ISI manages exact identification of the full set of HP causes for all Boolean model sizes considered in our experiments. Identifying the full set of causes took $0.01s \pm 0.02$ ($\sim 300 \pm 500$ calls), $2.5s \pm 1.7$ ($\sim 5e5 \pm 3e5$ calls), and $544s \pm 136$ ($\sim 4e6 \pm 1e6$ calls) for respectively 2, 5, and 10 attackers.

In the non-Boolean model, ISI "causally backtracks" to all canceling counterfactual interventions. In this case, the number of model calls is often comparable to that of MBS, but ISI consistently yields higher-quality cause sets.

Overall, replacing MBS with ISI increases the DICE score by up to 86.9 points (for $n = 5$ and $b = 4$ in the base model). In the Boolean model, ISI reduces the number of model calls by up to 99% (for $b = 256$ and $n = 2$) and always achieves at least a 40% reduction compared to MBS. In the non-Boolean model, ISI may require more calls than MBS (up to 68% more for $b = 16$ and $n = 10$), but for comparable quality levels, ISI remains more efficient. A full comparison between MBS and ISI is reported in Annex C.

## 6.3 Smallest cause identification

Figure 5 compares the runtime of the different algorithms (ILP and our methods with varying beam sizes) on the left, and the accuracy score of MBS on the right, as a function of model size (number of endogenous variables). We omit accuracy plots for ISI and ILP, as both achieve perfect accuracy across all tested configurations.

The results show that MBS performs well on small models but exhibits a drop in accuracy as the model size increases. Larger beam sizes delay this drop but do not eliminate it. On the runtime side, MBS is faster than ILP for small models but may become slower for larger models and large beam sizes.

In contrast, ISI maintains perfect DICE scores across all beam sizes and model sizes considered, while remaining significantly faster than ILP. These results highlight a limitation of MBS for smallest-cause identification and show that ISI combines the reliability of exact methods with substantially lower computational cost.

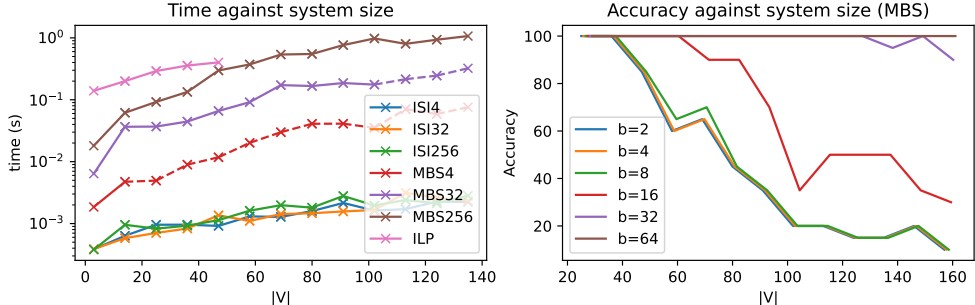

Figure 5: Time and accuracy vs. number of variables for smallest cause identification using ILP and various beam sizes for MBS and ISI. In the left figure, dotted lines are used to link points for which the accuracy is not 100%. The right figure only features the MBS algorithm, and the line colors do not share semantics with those of the left figure.

### 6.4  Impact of the LUCB algorithm in the stochastic model

Columns 3 and 4 of Figure 3 report results for stochastic models. Column 3 uses our LUCB algorithm for oracle and heuristic evaluation, while Column 4 uses naive average estimators. The DICE scores remain similar across both approaches, while the number of model calls is consistently lower with LUCB.

Quantitatively, LUCB reduces the number of model calls by up to 27%, with only minor losses in DICE score. We observe two cases of DICE degradation (for ISI with $n = 10$ and $b \leq 4$), while most configurations exhibit negligible differences (below 1%) or even improvements, with DICE gains of up to 11 points (for MBS with $n = 2$ and $b = 16$). A detailed comparison between LUCB and naive estimators is reported in Annex C.

## 7  Discussion

### 7.1  Limitations of the proposed algorithms

A key practical limitation of our approach is the requirement of an oracle function, which must support counterfactual generalization (Schölkopf et al., 2021), i.e., the ability to evaluate outcomes under unseen or even unrealistic interventions. Learning such a function from data remains beyond the capabilities of current machine learning models. In physical or numerical systems, implementing interventions can also be costly or infeasible. In practice, the most natural setting for our approach is simulation-based systems, where interventions and counterfactual evaluations are readily available. Many formal models of real-world systems can also be automatically translated into the SCM formalism (Ibrahim et al., 2020), further supporting the applicability of our method in these contexts.

We note, however, that this limitation should be understood relative to the requirements of HP causation itself. Any method grounded in the HP framework requires a fully specified SCM, including structural equations and a causal graph. Our oracle is strictly less demanding: it requires only the ability to evaluate counterfactual outcomes, without access to the structural equations. Moreover, when using MBS alone (without ISI), no causal graph is needed either. In this sense, the oracle assumption is a relaxation of the full SCM requirement rather than an additional constraint.

Another limitation concerns the design of the heuristic function. Our experiments in Annex E suggest that heuristics that minimize the number of positive values in the counterfactual state (such as "sum-pos") outperform neutral or counter-aligned ones such as heuristics that minimize the number of variables that remain at their actual values (such as "Occam"). These considerations are valid because the system is composed of conjunctions and disjunctions with few negations. They indicate that the choice of heuristic matters and that effective proxies can often be derived from the structure of the system. However, there are

no general guidelines for constructing effective heuristics. This can limit usability in practice. Nevertheless, in many domains, natural heuristics arise from the semantics of the system. For example, Reyd et al. (2025) use MBS to identify the cause of a flock of artificial agents colliding with an obstacle. In their simulation, agents that collide with the obstacle effectively merge with it; the distance to the obstacle's center therefore serves as a heuristic, as agents farther from the center are closer to avoiding the collision and thus canceling the target outcome. This example illustrates how domain knowledge can inform heuristic design without requiring problem-specific algorithmic changes.

Finally, our algorithms do not directly address continuous variable domains, which constitutes a major limitation. While discretization or sampling could be used in principle, this significantly reduces flexibility and may introduce additional approximation errors.

## 7.2 Limitations of the experimental evaluation

We acknowledge the limited scope of our experimental design. In particular, we do not compare against existing baselines for full HP-cause identification, as no general-purpose methods exist for this task. Similarly, our evaluation relies on a single family of SCMs. While this choice is motivated by the absence of established benchmarks, it necessarily limits the generality of our empirical conclusions. We mitigate this limitation through controlled variations of the model (Boolean, non-Boolean, and stochastic) and through sanity checks reported in Annex B.

The SMK benchmark also favors certain aspects of our approach, particularly ISI. In this model, the smallest cause always lies among the direct parents of the target variable, which explains the perfect accuracy observed when early stopping is used. In more complex SCMs, identifying the smallest cause may require less efficient approximations of ISI. Additionally, the non-Boolean variant of the SMK model bounds the size of non-minimal canceling interventions and restricts the number of values that cancel the outcome, limiting the combinatorial explosion that could otherwise arise.

The LUCB algorithm introduces several additional parameters, which may require tuning in practice. Though this adds practical concerns, it also provides explicit control over the precision–efficiency trade-off. A systematic study of the influence of these parameters is beyond the scope of this paper.

Finally, we report only a preliminary complexity analysis of MBS and ISI in Annex D. This complexity depends on properties of the identified causes, such as their number and size, which are difficult to characterize without carefully designed SCMs. A more comprehensive empirical complexity analysis is therefore left for future work.

## 7.3 Limitations of actual-cause definitions

The HP definition of actual causation has well-known conceptual limitations, including the lack of explicit consideration for responsibility (Chockler & Halpern, 2004), normality (Halpern & Hitchcock, 2015), and contrastiveness (Miller, 2021). In this paper, we focus on HP-causes because they constitute a widely accepted baseline and serve as the foundation for many alternative definitions.

Our algorithms assume that the first condition (AC1) is verified by the user and that the minimality condition (AC3) is approximately enforced by the search procedure. The second condition (AC2) is handled through the oracle function. While this reliance on an oracle is a limitation, it also provides flexibility: many extensions of the HP definition modify AC2, for instance to account for normality or causal sufficiency (Beckers, 2021). Treating the oracle as an input therefore allows our algorithms to adapt naturally to alternative definitions of actual causation, which is valuable given the lack of consensus on a definitive formalization.

## 7.4 Future work

Future work will address several directions. First, we will explore additional approximations of ISI, such as limiting subset sizes during "causal backtracking" or "causally backtracking" on HP-causes with non-Boolean systems. Second, we will investigate adaptations of our algorithms to continuous domains. Third, we aim to provide theoretical guarantees on approximation quality as a function of the heuristic and to

develop practical guidelines for heuristic design. Finally, and most importantly, we plan to integrate our algorithms into complete explanation frameworks, including causal discovery in real systems, oracle design aligned with refined notions of actual causation, and relevance-based filtering to produce human-interpretable explanations.

## 8 Conclusion

Actual causes play a central role in explainable artificial intelligence, as they align with users' expectations for explanations grounded in specific situations rather than general tendencies. Despite decades of work on formal definitions, identifying actual causes in practice remains challenging, and existing methods are scarce, restricted to narrow model classes, or limited to the smallest causes.

In this paper, we introduced the first practical algorithms for full HP-cause identification that extend beyond deterministic Boolean systems to non-Boolean and stochastic models. Our baseline algorithm provides an explicit and controllable trade-off between identification quality and computational cost. Building on this foundation, we proposed a complementary algorithm, which exploits structural properties of the system to significantly improve both accuracy and efficiency, and we formally proved its correctness. Finally, we introduced another complementary algorithm that improves sample efficiency in stochastic settings.

We evaluated our approach on a Boolean SCM from the literature and on non-Boolean and stochastic variants. Our results demonstrate that the beam size parameter effectively controls the quality–efficiency trade-off, that ISI consistently improves full cause identification compared to the baseline, that ISI outperforms existing approaches for smallest-cause identification when applicable, and that LUCB substantially reduces the number of model calls in stochastic models.

While our algorithms are not sufficient on their own to produce user-facing explanations and must be combined with complementary methods for relevance filtering and model construction, they address a major practical gap in the causality and XAI literature. By making actual cause identification computationally accessible across a broader class of systems, this work provides a concrete step toward more user-centric explanations and facilitates the broader adoption of actual causation in explainable AI research and applications.

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

# A  Full illustration of our algorithms

This appendix provides a pedagogical, step-by-step illustration of the operational behavior of our algorithms. It is intended to support intuition and clarify how the procedures unfold in practice; it is not required for understanding the theoretical contributions or the experimental results.

## A.1  MBS illustration on Example 1

We present a complete execution of the MBS algorithm on Example 1, schematized in Figure A.1. The causal graph is shown on the left-hand side of the figure. We run the algorithm without early stopping and without a maximum number of steps. The heuristic function $\psi$ minimizes the number of observed positive variables, and the beam size is set to $b = 3$. Each node in the search tree corresponds to an intervention, represented as a set of variable–value assignments. We use the compact notation introduced in Section 4.1, where a variable appears with or without the symbol $\neg$ depending on whether its assigned value.

**Step 1.**  The first level of the search tree includes all singleton interventions, each setting one variable to its counterfactual value. None of these interventions cancels the consequence. All nodes are therefore scored using $\psi$, and the three best ones are retained in the beam (shown in blue in Figure A.1). The remaining node is discarded (shown in red).

**Step 2.**  Each node in the beam is expanded. The expansion of $BH$ is shown with dotted lines to highlight that the algorithm does not build or evaluate these children. Generated nodes are evaluated; four interventions cancel the consequence and are identified as candidate causes (shown in green). Only the minimal ones are kept (dark green), while non-minimal causes (light green) are discarded. All nodes that are supersets of the identified causes are then pruned. The remaining nodes are scored, and the three best are retained.

**Step 3.**  The beam nodes are expanded again. Children that are supersets of already identified causes are not expanded further; these pruned expansions are shown in grey with dotted edges. After evaluation and scoring, the three best nodes are kept.

**Step 4.**  At this stage, two nodes are supersets of identified causes and are therefore ignored. The remaining two nodes are evaluated, but they already include all variables and cannot be expanded further. The algorithm terminates.

The final output consists of two causes of size 1, namely $ST$ and $SH$. For each cause, the algorithm also returns the associated witness set, which in both cases is $BH$.

## A.2  ISI algorithm illustration

We illustrate the behavior of ISI on the instance shown in Figure A.2 by first recalling how the base MBS algorithm behaves on this example, and then showing how ISI restructures the search to obtain the same causes more efficiently.

**Behavior of MBS on this instance**  As a baseline, we run MBS with a large beam size and a sufficiently large maximum number of steps so that no pruning is induced by these parameters. The heuristic $\psi$ again minimizes the number of positive variables.

MBS initializes the search with single-variable interventions and identifies one cause: $\{DK\}$. It expands the remaining candidates into interventions of size two, excluding $\{DK\}$. MBS then identifies $\{DK_2\}$, $\{GK_2\}$, and $\{GP_2\}$, all with witness $\{DK_3\}$ at step 2. Finally, it identifies $\{FS_2, FN_2\}$ and $\{FF_2, FDB_2\}$, again with witness $\{DK_3\}$, at step 3. MBS does not identify any cause during steps 4-6 and terminates.

This run illustrates that MBS is able to identify all HP-causes in this instance, but only after exploring a large portion of the intervention space.

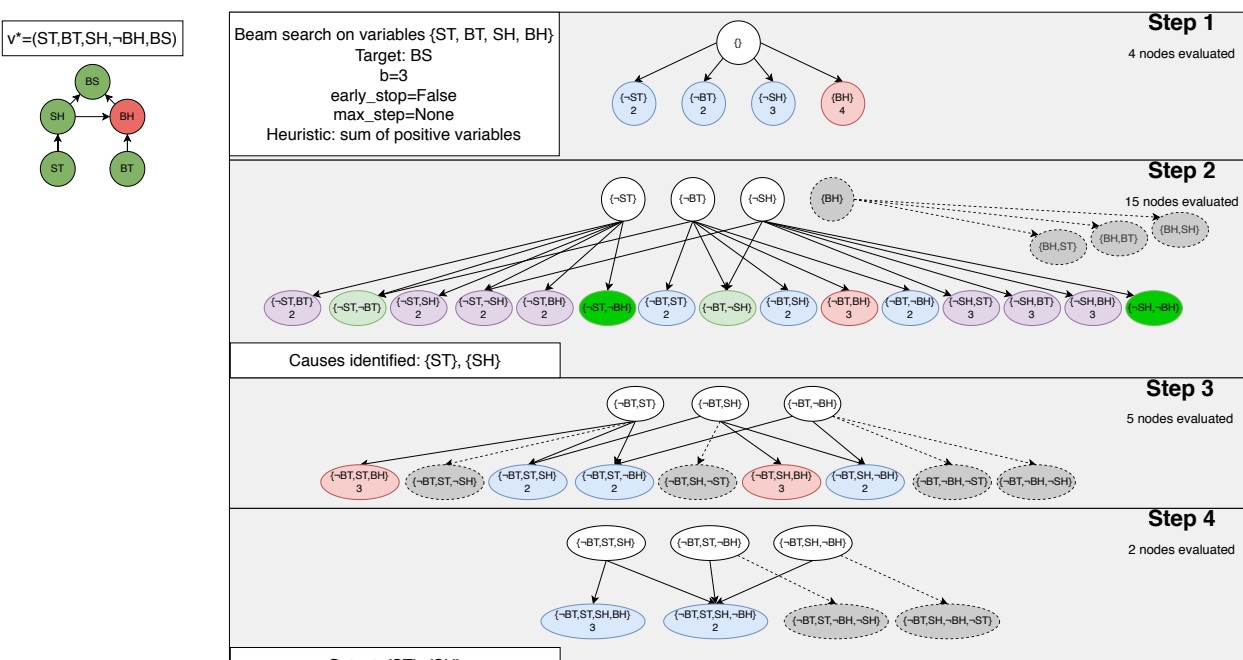

Figure A.1: Illustration of our algorithm on Example 1. Blue nodes constitute the beam, i.e., the nodes selected for expansion. Red nodes have less optimal heuristic values and are not part of the beam. Green nodes 'cancel' the consequence; light green ones are not minimal, and dark green ones are the identified causes. Purple nodes are evaluated, but filtered because they are supersets of a cause identified at the same depth. Nodes in gray (with dotted lines) are represented for illustration purposes, but are not evaluated.

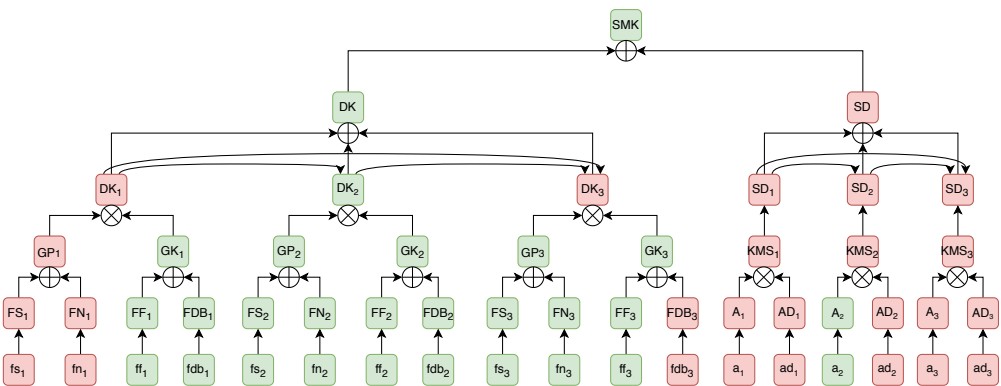

Figure A.2: An instance of the SMK scenario. Green nodes have value 1 and red ones have value 0. Nodes with lowercase letters are exogenous variables.

**Behavior of exact ISI on the same instance**   We now show how exact ISI decomposes this same search into structured subproblems. For clarity, we report only the inputs and outputs of each MBS call invoked by ISI, without detailing the internal beam-search steps.

*Step 1.* ISI starts by calling MBS on the parents of $SMK$, namely $\{DK, SD\}$. This call identifies the cause $\{DK\}$. ISI then enqueues the parents of $DK$, that is, $\{DK_1, DK_2, DK_3\}$.

*Step 2.* ISI runs MBS on $\{DK_1, DK_2, DK_3\}$ and identifies $\{DK_2\}$ with witness $\{DK_3\}$. ISI enqueues the parents of $DK_2$, namely $\{DK_1, GP_2, GK_2\}$, together with the witness.

*Step 3.* ISI runs MBS on $\{DK_1, GP_2, GK_2\}$ with witness $\{DK_3\}$ and identifies $\{GP_2\}$ and $\{GK_2\}$, both with witness $\{DK_3\}$. ISI enqueues their respective causal parents: $\{FS_2, FN_2\}$ and $\{FF_2, FDB_2\}$, again carrying over the same witness.

*Step 4.* ISI runs MBS on $\{FS_2, FN_2\}$ with witness $\{DK_3\}$ and identifies $\{FS_2, FN_2\}$ with witness $\{DK_3\}$. There are no children, so ISI does not enqueue anything.

*Step 5.* ISI runs MBS on $\{FF_2, FDB_2\}$ with witness $\{DK_3\}$ and identifies $\{FF_2, FDB_2\}$ with same witness. Once again, nothing is enqueued.

*Termination.* The queue is now empty. All relevant causal branches have been explored, and ISI terminates.

## B  Sanity check

This appendix reports sanity checks on classic examples from the actual causation literature. Specifically, we used the relevant examples used by Halpern (2015) as illustrations and means of analysis of definition 2. These examples are meant to illustrate the behavior of the actual cause definitions rather than to assess scalability or performance. Accordingly, we only aim to verify that the exact MBS and ISI generate the expected causes. Table 2 summarizes the expected causes and the outcomes of the identification process.

While this is not the objective of this section, we can still note that with these small models with flat structure, the ISI algorithm no longer seems faster than MBS (being even slower in some cases). ISI takes advantage of the structure of the graph to reduce the computational complexity; when this structure is not advantageous, or when there are so few variables that the computational complexity does not matter, ISI no longer outperforms MBS.

**Forest fire (Halpern & Pearl, 2001).**   This example contrasts conjunctive and disjunctive causation. A forest fire $FF$ may be caused by lightning $L$ or a dropped match $MD$. In the disjunctive case, $FF := L \lor MD$, both $\{L\}$ and $\{MD\}$ are expected causes when both occur. In the conjunctive case, $FF := L \land MD$, the only expected cause is $\{L, MD\}$. Halpern (2015) proposes an extended version introducing intermediary variables $A := L \land \neg MD$, $B := \neg L \land MD$, and $C := L \land MD$, with $FF := A \lor B \lor C$. In this model, $L$, $MD$, and $C$ are expected causes, each under appropriate witnesses.

| | n_calls | | t (ms) | | correct? | |
| | MBS | ISI | MBS | ISI | MBS | ISI |
| Model | | | | | | |
|---|---|---|---|---|---|---|
| Forest Fire Disjunctive | 5 | 5 | 0.35 | 0.14 | ✓ | ✓ |
| Forest Fire Conjunctive | 2 | 2 | 0.10 | 0.06 | ✓ | ✓ |
| Forest Fire Extended | 13 | 56 | 0.31 | 0.54 | ✓ | ✓ |
| Rock Throwing | 6 | 33 | 0.24 | 0.37 | ✓ | ✓ |
| Prisoners | 11 | 11 | 0.21 | 0.17 | ✓ | ✓ |
| Assassin | 5 | 5 | 0.14 | 0.10 | ✓ | ✓ |
| Lamp | 56 | 12 | 0.56 | 0.20 | ✓ | ✓ |
| Ranch | 96 | 96 | 1.52 | 1.03 | ✓ | ✓ |
| Ranch Extended | 623 | 3218 | 20.91 | 37.90 | ✓ | ✓ |
| Vote | 118 | 118 | 1.33 | 1.15 | ✓ | ✓ |
| Vote 3 Ways | 992 | 312 | 9.07 | 2.73 | ✓ | ✓ |

Table 2: Sanity check of MBS and ISI with an unlimited beam size against some classic small SCMs from the literature.

**Rock Throwing (Paul & Hall, 2013).**   This is Example 1 of the main paper. We recall only that the expected causes are Suzy throwing the rock $\{ST\}$ and Suzy hitting the bottle $\{SH\}$.

**Prisoners (Hopkins & Pearl, 2003).** This scenario illustrates how the HP definition excludes irrelevant variables. A prisoner dies if either $A$ loads $B$'s gun and $B$ shoots, or if $C$ loads and shoots his own gun: $D := (A \wedge B) \vee C$. In the context $a = c = 1$ and $b = 0$, the only expected cause is $\{C\}$, while $A$ is correctly excluded.

**Assassin (Hitchcock, 2007).** This scenario illustrates a case where the HP-definition conflicts with common intuitions and motivates the introduction of normality constraints. An assassin may poison a victim's drink ($AP$), while a bodyguard may add an antidote ($BA$), which neutralizes the poison and ensures the victim survives ($VS$). The model is given by $VS := \neg AP \vee BA$. We consider the context $ap = 0$ and $ba = 1$, where the assassin does nothing and the bodyguard adds the antidote. Intuitively, the victim's survival is expected and does not seem to have a meaningful cause. However, under the HP-definition, the set $AP, BA$ is identified as an actual cause of $VS$, highlighting the need for additional criteria such as normality to filter out such unintuitive causes.

**Lamp (Weslake, 2015).** This example is another illustration of how the HP definition excludes irrelevant variables. In this scenario, a lamp $L$ is controlled by three switches $A$, $B$, and $C$ that each have position $\{-1, 0, 1\}$. The lamp is on only if two switches share the same position. We consider the case where $A$ has position 1, while $B$ and $C$ have position $-1$. Here, $\mathcal{V} = \{A, B, C, L\}$, $\mathcal{U} = \{a, b, c\}$, $A := a$, $B := b$, $C := c$, $L := (A = B) \vee (A = C) \vee (B = C)$. The expected causes are $\{B\}$ and $\{C\}$.

**Ranch (Glymour et al., 2010).** This example models a group decision at a ranch, where five agents $A_1, \ldots, A_5$ vote on whether to leave a campfire for a roundup. The outcome $O$ follows a conditional protocol: if $A_1 = A_2$, their vote determines the outcome; if $A_2, \ldots, A_5$ agree while $A_1$ disagrees, $A_1$ decides; otherwise, the majority vote applies. In the context $a_1 = a_2 = 1$ and $a_3 = a_4 = a_5 = 0$, the first rule applies. The HP-causes are $\{A_1\}$ and all combinations involving $A_2$ and $A_3, \ldots, A_5$. Indeed, canceling $A_2$ alone would trigger the second rule, while canceling $A_2$ and another (besides $A_1$) would trigger rule 3 and cancel the consequence. Halpern (2015) introduces intermediary variables $M_1$, $M_2$, and $M_3$ to represent which decision rule is active, yielding the more intuitive causes $A_1$, $A_2$, and $M_1$.

**Vote (Livengood, 2013).** In a majority vote between two candidates with a 4-2 outcome, any pair of voters who voted for the winner constitutes a cause. Extending the model to three candidates with a 4-2-0 outcome preserves the same set of expected causes. We implement both cases with contexts $a_1 = a_2 = a_3 = a_4 = 1$ and $a_5 = a_6 = 0$.

## C    Measure of the impact of our algorithms

We report the decrease in the number of model calls (in %) and the increase in the quality of the identified cause set (in DICE points) when using the improved algorithms over their naive counterparts, i.e., ISI over Minimal Beam Search in Figure C.1 and LUCB over Naive Average in Figure C.2.

In the vast majority of settings, ISI reduces the number of model calls, with gains ranging from 33% to 99%. This trend breaks only in two types of settings, both in the non-Boolean model. First, for very small beam sizes ($b \leq 4$) with $n = 5$. Second, for most beam sizes ($4 \leq b \leq 128$) with $n = 10$. In these cases, ISI may call the model more often on average (up to 68% more). However, these settings also exhibit very high variance, which blurs this pattern. Moreover, the substantial gains in DICE observed in these cases suggest that using ISI remains beneficial despite the increased number of calls.

The gain in DICE points with ISI is systematic. For $n = 5$, ISI always increases the DICE score by more than 80 points. For $n = 10$, the improvement is nuanced for small beam sizes ($b \leq 16$), but represents more than 60 DICE points otherwise. For $n = 2$, the gains are less pronounced only because MBS performs well and ISI saturates at 100% DICE.

Similarly, LUCB reduces the number of model calls by 18% to 26% in nearly all settings. The only exception occurs for $n = 2$ when using MBS. The algorithm's execution explains this behavior: naive average estimates are sufficiently noisy that the algorithm sometimes adds incorrect causes and terminates one step too early,

| n | bs | Base Model | Non-Boolean |
|---|---|---|---|
| 2 | 2 | $75 \pm 43$ | $40 \pm 100$ |
| | 4 | $82 \pm 33$ | $52 \pm 82$ |
| | 8 | $89 \pm 22$ | $67 \pm 57$ |
| | 16 | $93 \pm 15$ | $78 \pm 40$ |
| | 32 | $95 \pm 9$ | $85 \pm 28$ |
| | 64 | $97 \pm 6$ | $89 \pm 21$ |
| | 128 | $98 \pm 3$ | $93 \pm 13$ |
| | 256 | $99 \pm 2$ | $96 \pm 8$ |
| 5 | 2 | $38 \pm 43$ | $5 \pm 62$ |
| | 4 | $50 \pm 31$ | $-26 \pm 111$ |
| | 8 | $59 \pm 25$ | $15 \pm 71$ |
| | 16 | $70 \pm 18$ | $32 \pm 47$ |
| | 32 | $78 \pm 14$ | $45 \pm 37$ |
| | 64 | $82 \pm 11$ | $58 \pm 26$ |
| | 128 | $86 \pm 10$ | $64 \pm 23$ |
| | 256 | $89 \pm 7$ | $70 \pm 19$ |
| 10 | 2 | $61 \pm 15$ | $34 \pm 36$ |
| | 4 | $59 \pm 14$ | $-2 \pm 91$ |
| | 8 | $58 \pm 16$ | $-32 \pm 98$ |
| | 16 | $58 \pm 15$ | $-68 \pm 229$ |
| | 32 | $60 \pm 14$ | $-52 \pm 132$ |
| | 64 | $62 \pm 11$ | $-12 \pm 80$ |
| | 128 | $63 \pm 10$ | $-2 \pm 62$ |
| | 256 | $65 \pm 8$ | $33 \pm 17$ |

(a) % of calls gained using ISI

| n | bs | Base Model | Non-Boolean |
|---|---|---|---|
| 2 | 2 | $37 \pm 21$ | $35 \pm 27$ |
| | 4 | $26 \pm 25$ | $26 \pm 27$ |
| | 8 | $19 \pm 24$ | $19 \pm 25$ |
| | 16 | $0 \pm 0$ | $0 \pm 0$ |
| | 32 | $0 \pm 0$ | $0 \pm 0$ |
| | 64 | $0 \pm 0$ | $0 \pm 0$ |
| | 128 | $0 \pm 0$ | $0 \pm 0$ |
| | 256 | $0 \pm 0$ | $0 \pm 0$ |
| 5 | 2 | $83 \pm 17$ | $83 \pm 17$ |
| | 4 | $87 \pm 7$ | $87 \pm 6$ |
| | 8 | $88 \pm 8$ | $89 \pm 7$ |
| | 16 | $85 \pm 8$ | $88 \pm 7$ |
| | 32 | $86 \pm 9$ | $85 \pm 8$ |
| | 64 | $87 \pm 10$ | $87 \pm 10$ |
| | 128 | $88 \pm 14$ | $88 \pm 12$ |
| | 256 | $82 \pm 25$ | $88 \pm 17$ |
| 10 | 2 | $22 \pm 21$ | $21 \pm 21$ |
| | 4 | $32 \pm 26$ | $32 \pm 27$ |
| | 8 | $44 \pm 31$ | $44 \pm 32$ |
| | 16 | $54 \pm 36$ | $55 \pm 36$ |
| | 32 | $60 \pm 36$ | $65 \pm 36$ |
| | 64 | $72 \pm 31$ | $71 \pm 31$ |
| | 128 | $79 \pm 24$ | $79 \pm 25$ |
| | 256 | $87 \pm 8$ | $86 \pm 9$ |

(b) Dice points gained using ISI

Figure C.1: Performance improvement due to ISI

| n | bs | Beam Search | ISI |
|---|---|---|---|
| 2 | 2 | -20 ± 19 | 19 ± 10 |
|  | 4 | -20 ± 15 | 18 ± 11 |
|  | 8 | -19 ± 14 | 17 ± 12 |
|  | 16 | -26 ± 14 | 16 ± 13 |
|  | 32 | -38 ± 22 | 16 ± 14 |
|  | 64 | -51 ± 29 | 16 ± 14 |
|  | 128 | -30 ± 37 | 16 ± 14 |
|  | 256 | 11 ± 27 | 16 ± 14 |
| 5 | 2 | 19 ± 5 | 18 ± 6 |
|  | 4 | 19 ± 4 | 18 ± 5 |
|  | 8 | 20 ± 0 | 17 ± 6 |
|  | 16 | 20 ± 0 | 18 ± 6 |
|  | 32 | 20 ± 0 | 18 ± 5 |
|  | 64 | 20 ± 0 | 18 ± 5 |
|  | 128 | 20 ± 0 | 18 ± 4 |
|  | 256 | 20 ± 0 | 18 ± 3 |
| 10 | 2 | 20 ± 1 | 25 ± 15 |
|  | 4 | 20 ± 0 | 26 ± 8 |
|  | 8 | 20 ± 0 | 23 ± 9 |
|  | 16 | 20 ± 0 | 21 ± 5 |
|  | 32 | 20 ± 0 | 20 ± 3 |
|  | 64 | 20 ± 0 | 19 ± 2 |
|  | 128 | 20 ± 0 | 19 ± 2 |
|  | 256 | 20 ± 0 | 19 ± 1 |

(a) % of calls gained using LUCB

| n | bs | Beam Search | ISI |
|---|---|---|---|
| 2 | 2 | 4 ± 6 | 4 ± 10 |
|  | 4 | 6 ± 5 | 4 ± 9 |
|  | 8 | 9 ± 9 | 5 ± 10 |
|  | 16 | 11 ± 12 | 5 ± 10 |
|  | 32 | 6 ± 12 | 5 ± 10 |
|  | 64 | 4 ± 12 | 4 ± 10 |
|  | 128 | 3 ± 9 | 4 ± 10 |
|  | 256 | 9 ± 7 | 4 ± 10 |
| 5 | 2 | 0 ± 1 | -1 ± 4 |
|  | 4 | -0 ± 1 | 0 ± 0 |
|  | 8 | 0 ± 1 | 0 ± 0 |
|  | 16 | -0 ± 1 | 0 ± 0 |
|  | 32 | 0 ± 1 | 0 ± 0 |
|  | 64 | -0 ± 1 | 0 ± 0 |
|  | 128 | -0 ± 1 | 0 ± 0 |
|  | 256 | 0 ± 2 | 0 ± 0 |
| 10 | 2 | 0 ± 0 | -4 ± 13 |
|  | 4 | -0 ± 0 | -5 ± 10 |
|  | 8 | 0 ± 0 | 0 ± 12 |
|  | 16 | 0 ± 0 | 1 ± 8 |
|  | 32 | 0 ± 1 | 2 ± 12 |
|  | 64 | 0 ± 1 | 4 ± 7 |
|  | 128 | -0 ± 1 | 4 ± 5 |
|  | 256 | -0 ± 1 | 4 ± 5 |

(b) Dice points gained using LUCB

Figure C.2: Performance improvement due to LUCB

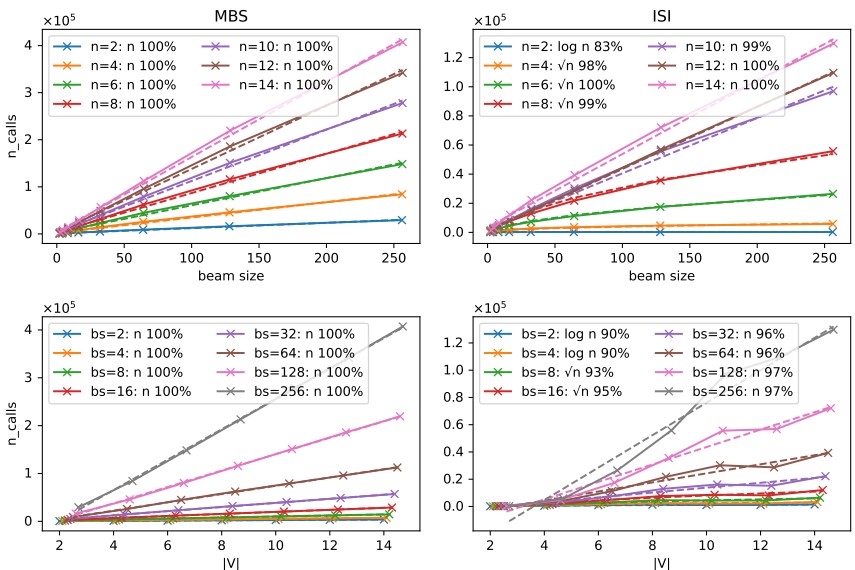

Figure D.1: Time complexity against the number of attackers and beam size for the full identification.

which reduces the overall number of model calls, even though LUCB reduces the number of calls at each step.

In most settings, LUCB either improves the DICE score or limits its degradation to less than one point. A notable exception occurs for $n = 10$, small beam sizes ($b \leq 4$), and ISI, where a more significant loss in DICE occurs. However, such low beam sizes are not recommended in practice, and we consider these cases to be non-representative edge cases rather than indicative of the algorithm's typical behavior.

# D   Complexity analysis

This annex provides a preliminary analysis of the complexity of our algorithms. We conduct a theoretical analysis followed by an empirical one. They both rely on assumptions based on the model used. These results are illustrative and provide early intuition into the complexity of our algorithms; they are not technical contributions.

## D.1   Theoretical analysis

The MBS algorithm performs $K \leq |\mathcal{V}|$ iteration steps. At each step $k$, it evaluates the current intervention set $E_k$ whose size is bounded by the beam size $b$ times the number of size-one interventions $|\mathcal{V}| \times |\mathrm{Dom}_m|$, where $\mathrm{Dom}_m$ is the largest domain. Then, MBS performs a minimality merge with the already identified causes $\mathbf{C}$, which requires iterations bounded by the size of the largest cause $C_m$. The number of operations is bounded by:

$$O(K \times b \times |\mathcal{V}| \times |\mathrm{Dom}_m| \times |\mathbf{C}| \times |C_m|)$$

We can further simplify this expression depending on the system or on the usage of MBS. Notably, $K$ is either bounded by $|\mathcal{V}|$ when no maximum number of steps is given to the algorithm or bounded by the provided parameter. $|C_m|$ highly depends on the system nature, but is notably bounded either by $|\mathcal{V}|$ or by the maximum number when provided. Concerning $|\mathbf{C}|$, the only formal bound is $2^{|\mathcal{V}|}$. However, in practice, it can be bounded by the system nature or by the MBS approximations. Notably, in the SMK scenario, both $|\mathbf{C}|$ and $|C_m|$ are bounded regardless of the system size (as only one attacker can cause $DK$ or $SD$). Hence, if we suppose $|\mathbf{C}|$ is bounded, we can expect a complexity linear in $|\mathrm{Dom}_m|$, linear in $b$ and either linear,

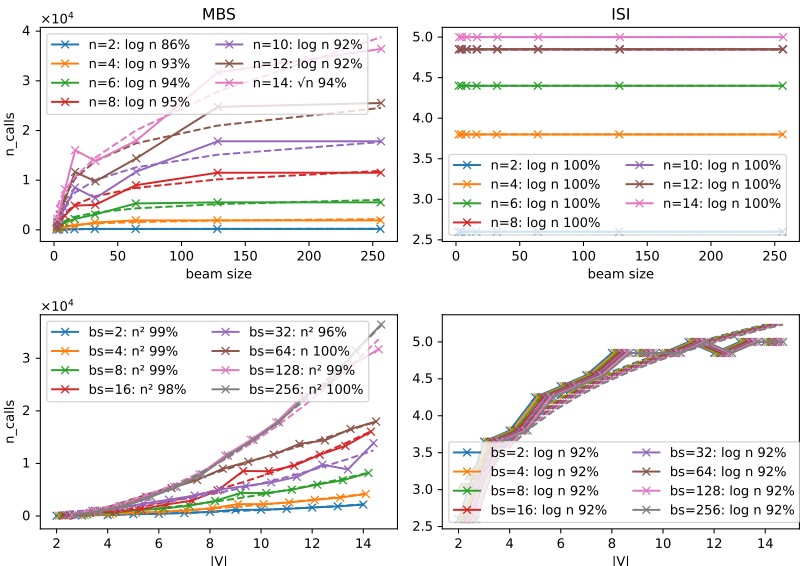

Figure D.2: Time complexity against the number of attackers and beam size for the smallest identification.

quadratic, or cubic in $|\mathcal{V}|$. We can expect $O(|\mathcal{V}|)$ if we impose a maximum number of steps, and otherwise $O(|\mathcal{V}|^2)$ if the size of the causes is naturally bounded by the system's nature or $O(|\mathcal{V}|^3)$ if not.

For the smallest cause identification, we stop when we find a cause, so we never iterate over causes or inside causes. Hence, we can expect linearity with respect to $b$ and $|\mathrm{Dom}_m|$, and either linear or quadratic dependence in $|\mathcal{V}|$ depending on whether we use `max_steps`.

ISI enqueues all subsets of all "canceling" interventions and runs the MBS algorithm on instances bounded by the larger number of parents possible for a "canceling" intervention $\mathrm{Pa}_m$. We must combine the complexity of ISI ($2^{|C_m|} \times |\mathbf{C}|$, where $\mathbf{C}$ are HP-causes for boolean ISI and "canceling" interventions otherwise) and the complexity of MBS called on a subinstance. The number of operations is therefore bounded by:

$$2^{|C_m|} \times |\mathbf{C}|^2 \times b \times |\mathrm{Pa}_m| \times |\mathrm{Dom}_m|$$

The exponential complexity is concerning, but disappears when the system's nature bounds the size of the larger cause.

In this paper, to identify the smallest cause with ISI, we limit ourselves to the direct parents of $T$. Hence, we expect a $O(1)$ complexity, which would not be the case using other ISI approximations or more challenging models.

## D.2 Empirical analysis

We approximate the empirical complexity of our algorithm in terms of the number of variables and beam size. We run regression for several complexities, i.e., we perform a linear regression between a function of the evaluated parameter ($b$ or $|\mathcal{V}|$) and the number of model calls. We test $O(\log(\cdot))$, $O(\sqrt{\cdot})$, $O(\cdot)$, $O(\cdot^2)$, and $O(\cdot^3)$. We then measure the $R^2$ coefficient for each linear regression and report the best one in % rounded to the nearest integer.

Figure D.1 shows the regressions for the full identification task. MBS scales linearly with both the beam size and the model size. This confirms our analysis as we used a fixed `max_steps` parameter. ISI almost scales with $\sqrt{b}$ for low values of $n$ and linearly for large values of $n$. Additionally, for $n = 2$, the closer complexity is $O(\log(\cdot))$. Overall, this suggests that the complexity remains less than linear but depends on $n$. Finally,

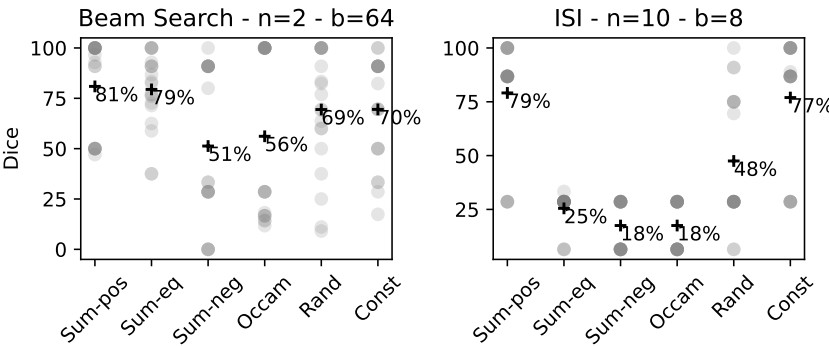

Figure E.1: Comparing DICE scores for various heuristics. Dots are the scores for each run, crosses are the averaged values.

ISI does not match any of our test complexities in terms of model size, but roughly scales similarly to $b$. The best fit is $O(\log(|\mathcal{V}|))$ for low $b$, $O(\sqrt{|\mathcal{V}|})$ for medium $b$, and $O(|\mathcal{V}|)$ for large $b$.

Figure D.2 shows the regressions for the smallest identification task. First, MBS scales quadratically with $b$ as expected, as no maximum number of steps is imposed. In terms of beam size, the closest complexity is almost always $O(\log(b))$, which is lower than the expected linear complexity. ISI does not depend on $b$ and is closest to $O(\log(|V|))$. This smooth increase (instead of the expected $O(1)$ complexity) is probably due to the probability of a cause of size 2 being larger in larger models. Hence, the number of MBS runs with two steps increases compared to those with one step when $|V|$ increases.

## E  Comparison of heuristics

This annex provides illustrative heuristic functions for our SMK model and shows that they influence the algorithm's performance, mostly as expected based on their design. Section 7.1 discusses the practical challenge of designing heuristics for real-world systems and gives a concrete example of how domain knowledge can guide this choice.

We present six heuristics based on the model counterfactual states that are to be minimized. First, *"sum-pos"*, which we used for the main paper content experiments, computes the sum of positive values. We expect it to perform well as our model uses "and" and "or" based logical formulas with very few "not". Second, we use *"sum-eq"*, which computes the number of counterfactual values observed in a counterfactual state. This heuristic favors large interventions that apply to variables "far" from $T$. Then, we use *"sum-neq"*, the complement of "sum-pos", which counts the number of variables with value 0. We expect this heuristic to yield poor results. Then, we use the complement of "sum-eq", which counts the number of variables that remain at their actual values. We call this heuristic *"Occam,"* as it can be seen as a measure of how simple it is to describe the counterfactual state compared to the actual one. This measure can be used to assess the quality of causes, as simple ones will provide better explanations (Miller, 2019; Kahneman & Miller, 1986). However, we do not expect this metric to yield good results as it favors interventions with little effect on the model, which likely does not imply that we are close to "canceling" $T$. Finally, we use two witness heuristics: the *"random"* heuristic that returns a random integer between 0 and $|V|$ and the *"constant"* heuristic that always returns $|V|/2$.

Figure E.1 shows the DICE scores obtained with our heuristics in the Boolean model, first with MBS, $n = 2$ and $b = 64$, second with ISI, $n = 10$ and $b = 8$. These settings correspond to medium DICE values with our original heuristic. In the first case, "sum-pos" and "sum-eq" both perform well and beat the two witness heuristics, contrary to "sum-neg" and "Occam". In the second case, "sum-pos" is still the best heuristic, but the "constant" heuristic almost reaches the same DICE score (79% against 77%). As we could expect, the "random" heuristic yields worse results, and "sum-neg" and "Occam" even worse ones. Quite unexpectedly, "sum-eq" also yields worse results than "random".

The neutral heuristics ("random" and "constant") outperforming the counter-aligned ones ("sum-neg" and "Occam") illustrates that carrying no signal is preferable to actively pointing away from the cancellation goal. This also explains why "Occam", despite being a natural measure of explanation quality, performs poorly as a search heuristic: minimizing the footprint of an intervention is desirable for explanations but counter-productive for finding interventions that cancel $T$.

## F    Proofs of theorems

This section formalizes notions introduced in Section 4.2, introduces additional notions necessary to prove theorems 1 and 2, and presents the proof of all lemmas and theorems. Contrary to other annexes, this annex constitutes a technical contribution and is not in the main content for readability purposes only.

At a high level, the proofs show that all HP-causes can be reached by recursively exploring subproblems generated from the target's parents. As presented in Section 4.2, our algorithm recursively performs a "causal backtracking" operation on the "canceling interventions" identified in a given subproblem. The main challenge of the proofs is to explicitly characterize what our "causal backtracking" identifies.

Both proofs follow a similar induction pattern. We define a measure of "depth" of a "canceling intervention", and we proceed by induction. We first show that the "canceling interventions" with the "smallest depth" are in the parents of the target. We then suppose that we identify all "canceling interventions" up to a certain "depth" and consider some "canceling interventions" $C'$ at the "next depth". We construct some "canceling interventions" $C$ at a "depth" that ensures that $C$ has been identified. We finally show that we can perform our "causal backtracking" from $C$ to identify $C'$. Sections F.2 and F.3 notably specify what exactly is identified by this "causal backtracking" and what notion of "depth" we use for each proof.

We do not aim for our proofs to be so formal that they can be automatically verified. However, we introduce numerous formal definitions mostly for clarity, as we hope they help remove ambiguities. We will only consider models with a finite set of variables with finite domains. We will stick to the convention that endogenous variables and sets of endogenous variables are denoted using capital letters. We will suppose throughout this section a fixed SCM $\mathcal{M} = (\mathcal{V}, \mathcal{U}, \mathrm{F}, \mathrm{Dom})$ together with a context $u \in \mathcal{U}$ and a target $T \in \mathcal{V}$. We will use, without an introduction, the set of their actual values (under the implicit model and context) as the same letter with an asterisk. For instance, the actual value of a variable $X \in \mathcal{V}$ will always be denoted $x^*$. Additionally, given a set of variables $S \subseteq \mathcal{V}$, with some values $s \in \mathrm{Dom}(S)$, we will refer to the restriction of these values to the subset $S' \subseteq S$ as $s_{|S'} \in \mathrm{Dom}(S')$. Finally, we will assume that $\mathcal{V} = \mathrm{An}(T)$. This can be done without loss of generality since the variables that are not ancestors of $T$ have no influence on it and don't change the set of causes. Additionally, reducing a DAG to the set of ancestors of $T$ is immediate.

### F.1    Formalisation of Section 4.2

We first formalize the notion mentioned or implied in Section 4.2. Notably, we regularly mention "canceling" interventions, or the solution to the subproblems induced by some $\kappa = (I, W_0, R_C, r_C, R_W)$. We will also formally introduce the procedure that generates "subproblems" based on the solution of a given "subproblem", the corresponding recursive operator, and its closure.

**Definition 3 (Set of HP-causes)** *We denote $\mathcal{C}$ the set of all sets of variables $C \subseteq \mathcal{V}$ that qualify as HP-causes of a target $T$ in a model and context $(\mathcal{M}, u)$, i.e., that satisfy AC1, AC2 and AC3 from definition 2.*

**Definition 4 ("Canceling" interventions)** *We call "canceling" interventions the set $\mathcal{A}_{\mathrm{AC2}}$ of all pairs $(C, c)$ that satisfy AC2.*

$$\mathcal{A}_{\mathrm{AC2}} = \{(C, c) | C \subseteq \mathcal{V}, c \in \mathrm{Dom}(C), \forall X \in C : c_{|\{X\}} \neq x^*,$$
$$\exists W \subseteq \mathcal{V} \setminus C : (\mathcal{M}, u) \models [C \leftarrow c, W \leftarrow w^*](\neg T)\}$$

**Definition 5 (Witness set)** *We define the set $\mathcal{W}$ of all witnesses of a pair $(C, c) \in \mathcal{A}_{\mathrm{AC2}}$.*

$$\mathcal{W}(C, c) = \{W \subseteq \mathcal{V} | W \cap C = \emptyset, (\mathcal{M}, u) \models [C \leftarrow c, W \leftarrow w^*](\neg T)\}$$

*Notably, we can rewrite Definition 4 as $\mathcal{A}_{\text{AC2}} = \{(C, c) | C \subseteq \mathcal{V}, c \in \text{Dom}(C), \forall X \in C : c_{|\{X\}} \neq x^*, \mathcal{W}(C, c) \neq \emptyset\}$*

**Definition 6 (Constrained witness)** *Given disjoint sets of variables $I, R \in \mathcal{V}$, where $R$ is called the reference set and $I$ is called the free instance, we define the set of constrained witnesses $\mathcal{W}_{R,I}$ as all the witnesses of a pair $(C, c) \in \mathcal{A}_{\text{AC2}}$ that include $R$ and are included in $I \cup R$:*

$$\mathcal{W}_{R,I}(C, c) = \{W \subseteq \mathcal{W}(C, c) | R \subseteq W \subseteq I \cup R\}$$

**Definition 7 (AC2 constraining sets)** *We call AC2 constraining sets $\kappa = (I, W_0, R_C, r_C, R_W)$ the disjoint sets $I, W_0, R_C, R_W \subseteq \mathcal{V}$ and the values $r_C \in \text{Dom}(R_C)$, where $I$ is the free instance, $R_C$ and $R_W$ are the set of restricted variables, where variables in $R_C$ are forced to values in $r_C$ and variables in $R_W$ are forced to their actual values $r_W^*$, and $W_0$ the set of optional witness variables.*

**Definition 8 (Constrained "canceling" interventions)** *Given a constraining set $\kappa = (I, W_0, R_C, r_C, R_W)$, we define $\mathcal{A}_{\text{AC2}}[\kappa]$ as a set of pairs $(C, c)$ that satisfy AC2 and are constrained by $\kappa$.*

$$(C, c) \in \mathcal{A}_{\text{AC2}}[\kappa] \subseteq \mathcal{A}_{\text{AC2}} \text{ if}$$

$$\mathcal{W}_{R_W, I \cup W_0}(C, c) \neq \emptyset \qquad R_C \subseteq C \subseteq R \cup I \qquad c_{|R_C} = r_C$$

*The set of all "canceling" interventions without any constraint is then $\mathcal{A}_{\text{AC2}} = \mathcal{A}_{\text{AC2}}\big[(\mathcal{V}, \emptyset, \emptyset, \emptyset, \emptyset)\big]$.*

**Definition 9 (Constrained HP-causes)** *Given a constraining set $\kappa = (I, W_0, R_C, r_C, R_W)$, we define $\mathcal{C}[\kappa]$ as a set of sets of variables $C \subseteq \mathcal{V}$ that satisfy AC2, are minimal, and are constrained by $\kappa$.*

$$C \in \mathcal{C}[\kappa] \text{ if}$$

$$\exists c \in \text{Dom}(C) : \mathcal{W}_{R_W, I \cup W_0}(C, c) \neq \emptyset \qquad R_C \subseteq C \subseteq R \cup I \qquad \forall C' \in \mathcal{C}[\kappa] : C' \not\subseteq C$$

*The set of all HP causes without any constraint is then $\mathcal{C} = \mathcal{C}\big[(\mathcal{V}, \emptyset, \emptyset, \emptyset, \emptyset)\big]$.*

**Definition 10 (Counterfactual Expansion (CF-expansion))** *Given a constraining set $\kappa = (I, W_0, R_C, r_C, R_W)$, the CF-expansion operator $cfx$ generates a set of new constraining sets $\kappa' = (I', W_0', R_C', r_C', R_W')$ by iterating over the "canceling" CF-interventions $(C, c)$ constrained by $\kappa$ and over the non-empty subsets $S$ of $C$.*

$$cfx(\kappa) = \{\kappa' = (I', W_0', R_C', r_C', R_W') \mid S \in 2^C \setminus \emptyset, \ (C, c) \in \mathcal{A}_{\text{AC2}}[\kappa]\}$$

*where:*

$$I' = \text{Pa}(S) \setminus C \qquad W_0' = \text{De}^-(I') \cap \text{An}^-(C)$$

$$R_C' = (C \setminus S) \qquad r_C' = c_{|R_C'} \qquad R_W' = W \cup \text{De}^{-\setminus C}(I')$$

**Definition 11 (CF-Expansion Operator)** *Given a set $K$ of constraining sets $\kappa$, we define the CF-Expansion Operator $\mathcal{CFX}$ that generates the set of all constraining sets obtained by recursively calling $cfx$ on its outputs.*

$$\mathcal{CFX}^0(K) = K \qquad \mathcal{CFX}^k(K) = \bigcup_{\kappa \in \mathcal{CFX}^{k-1}(K)} cfx(\kappa) \qquad \mathcal{CFX}^*(K) = \bigcup_{k=0}^{\infty} \mathcal{CFX}^k(K)$$

*$\mathcal{CFX}^*$ can notably be seen as the fixed point of the operator. Since $\mathcal{CFX}$ takes its values in a finite set, $\mathcal{CFX}^*$ is well defined.*

**Definition 12 (Counterfactual Closure of the Target)** *Given the target $T$, we call counterfactual closure of the target (CF-closure of $T$) the set $\mathcal{CFX}^*(\{\kappa_0\})$, with $\kappa_0 = (\mathrm{Pa}(T), \emptyset, \emptyset, \emptyset, \emptyset)$.*

**Remark 3 (Boolean CF-Expansion)** *We suppose boolean analogous to the CF-expansion procedure $cfx_b$, to the CF-expansion operator $\mathcal{CFX}_b$, and to the CF-closure of $T$ $\mathcal{CFX}_b(\{\kappa_0\})$. The boolean CF-expansion is similar to the original CF-expansion but builds upon constrained HP-causes $\mathcal{C}[\kappa]$ instead of more general "canceling" interventions $\mathcal{A}_{\mathrm{AC2}}[\kappa]$. Additionally, it uses $r'_C = \neg c^*_{|R_C}$ as counterfactual intervention values in the constraining set. We do not provide further formal definition as we deem this description exhaustive enough.*

We now introduce a general lemma that will be useful for our proofs.

**Lemma 4 (Witness can be taken inside descendants)** *Let $C$ be the variables of a "canceling" intervention (i.e., it satisfies AC2) with some witness $W$.*

*Then there exists $W' \subseteq \mathrm{De}(C)$ such that $W'$ is a witness for $C$ (i.e., AC2 still holds when replacing $W$ by $W'$).*

*In the remainder of this section, we will always assume that $W \subseteq \mathrm{De}(C)$ when $W$ is a witness for cause $C$, i.e., $W \in \mathcal{W}(C, \cdot)$.*

**Proof** *Let $C, W \subseteq \mathcal{V}$ be an HP cause and its witness. Partition $W$ into $W_A = W \cap \mathrm{De}(C)$ and $W_B = W \setminus \mathrm{De}(C)$. Then $W_A$ is a witness for $C$.*

*Indeed, an intervention on some variables can only influence the values of their descendant. Hence, $W_B$ cannot be influenced by interventions on $C$ or $W_A$ and naturally takes values $w^*_B$ under intervention $[C \leftarrow c, W_A \leftarrow w^*_A]$. Hence, this counterfactual model is the same with and without intervention $[W_B \leftarrow w^*_B]$. Therefore, we have $(\mathcal{M}, u) \models [C \leftarrow c, W_A \leftarrow w^*_A](\neg T)$, so $W_A$ is a witness for $C$.* ∎

### F.2 Proof of theorem 1

This subsection introduces additional notions necessary to prove Theorem 1 and proceeds to prove the introduced results as well as Theorem 1 itself.

This subsection aims to prove that all HP-causes can be identified by performing a recursive "causal backtracking" on the already identified "canceling interventions", starting from the target's parents. As we saw at the beginning of this section, the idea of the proof is to perform an induction on the "depth" of the "canceling interventions". We suppose that we have identified all causes up to a certain "depth" $n$ and we consider some cause $C'$ at depth $n+1$, build a cause $C$ at depth $n$, and show that we can "causally backtrack" from $C$ to $C'$. The main challenge of this proof is that ISI does not necessarily identify all "canceling interventions" but only the "meaningful ones".

In this subsection, we define the notion of depth that will be used for the induction (Definition 13), the notion of "meaningful canceling intervention" (Definition 15) that is actually what ISI identifies by "causally backtracking" on "canceling interventions" (Lemma 5). We finally prove that the HP causes are all "meaningful canceling interventions" (Lemma 5), which proves that ISI identifies all HP-causes (Theorem 1).

**Definition 13 (Depth)** *For a node $X \in \mathcal{V}$ or a set of nodes $S \subseteq \mathcal{V}$, we define the depth of $X$, $d(X)$, as the size of the longest path from $X$ or $S$ to the target. Since there are no cycles in a DAG and $\mathcal{V} = \mathrm{An}(T)$, this is well defined. We can formalize this definition as such:*

$$\forall S \subseteq \mathcal{V} : d(S) = \max_{Y \in S} d(Y) \qquad \forall X \in \mathcal{V} : d(X) = \begin{cases} 1 \text{ if } \mathrm{Ch}(X) = \{T\} \\ 1 + d(\mathrm{Ch}(X)) \text{ otherwise} \end{cases}$$

**Definition 14 (Counterfactual-active (CF-active) path)** *Given a "canceling" intervention $(C, c)$, a witness $W \in \mathcal{W}(C, c)$ and a variable $X \in C$, we define the set of CF-active paths $\mathcal{P}_{\mathrm{cf}}(X; C, c, W)$ as the*

set of paths from $X$ to $T$ where all variables take counterfactual values under interventions $[C \leftarrow c]$ and $[W \leftarrow w^*]$. Formally, this means that:

$$\forall P \in \mathcal{P}_{\text{cf}}(X; C, c, W), \forall Y \in P : (\mathcal{M}, u) \models (C \leftarrow c, W \leftarrow w^*)[Y \neq y^*]$$

**Definition 15 (CF-active "canceling" interventions)** *We define the set of CF-active "canceling" interventions $\mathcal{A}_{\text{AC2}}^{\text{cf}}$ as the set of "canceling" interventions $(C, c)$ where all variables $X \in C$ have a non-empty set of CF-active path: $\exists W \in \mathcal{W}(C, c) : \mathcal{P}_{\text{cf}}(X; C, c, W) \neq \emptyset$.*

*We extend the notion of CF-active "canceling" interventions with constraining sets $\kappa$.*

$$\mathcal{A}_{\text{AC2}}^{\text{cf}}[\kappa] = \mathcal{A}_{\text{AC2}}^{\text{cf}} \cap \mathcal{A}_{\text{AC2}}[\kappa]$$

**Lemma 5 (HP causes are CF-active AC2 pairs)** *All HP-causes are part of a CF-active "canceling" intervention.*

$$\forall C \in \mathcal{C} : \exists c \in \text{Dom}(C), (C, c) \in \mathcal{A}_{\text{AC2}}^{\text{cf}}$$

**Proof** *Let $C$ be an HP-cause of $T$. Then there is a setting $c$ and a witness $W$ such that AC2 is verified. Additionally, AC3 is also satisfied, meaning that no subset of $C$ satisfies AC2. Let $\xi = [C \leftarrow c, W \leftarrow w^*]$ be the intervention that "cancels" $T$, i.e., $(\mathcal{M}, u) \models [\xi](\neg T)$*

*Let $X \in C$. By way of contradiction, suppose that there is no CF-active path from $X$ to $T$. Then, for any path from $X$ to $T$, there is a variable that takes its actual value under $[C \leftarrow c, W \leftarrow w^*]$. Let $W'$ be the set of descendant of $X$ that take their actual values under interventions $\xi$. Under the intervention where $X$ is removed from $C$ and $W'$ is added with actual values, i.e., $[C \setminus \{X\} \leftarrow c|_{C \setminus \{X\}}, W \leftarrow w^*, W' \leftarrow w'^*]$, every path from $X$ to $T$ is cut by a node fixed to its actual value, hence $T$ remains false. Therefore AC2 still holds for $C \setminus \{X\}$ with witness $W \cup W'$, contradicting AC3.* ∎

We now introduce the most important lemma for proving Theorem 1. Most of this subsection's work lies in following the previous lemma.

**Lemma 6 ("Range" of $\mathcal{CFX}^*(\{\kappa_0\})$)** *All CF-active "canceling" interventions $(C, c) \in \mathcal{A}_{\text{AC2}}^{\text{cf}}$ are constrained "canceling" interventions $(C, c) \in \mathcal{A}_{\text{AC2}}^{\text{cf}}[\kappa]$ with constraining set $\kappa \in \mathcal{CFX}^*(\{\kappa_0\})$.*

$$\mathcal{A}_{\text{AC2}}^{\text{cf}} \subseteq \bigcup_{\kappa \in \mathcal{CFX}^*(\{\kappa_0\})} \mathcal{A}_{\text{AC2}}^{\text{cf}}[\kappa]$$

**Proof** *We proceed by induction over the depth of "canceling" interventions, i.e., $n = d(C)$ for $(C, c) \in \mathcal{A}_{\text{AC2}}^{\text{cf}}$. For easier notation, we denote $\bigcup(...) \equiv \bigcup_{\kappa \in \mathcal{CFX}^*(\{\kappa_0\})} \mathcal{A}_{\text{AC2}}^{\text{cf}}[\kappa]$. We aim to prove that $\mathcal{A}_{\text{AC2}}^{\text{cf}} \subseteq \bigcup(...)$.*

**Initialization.** *Let $(C, c) \in \mathcal{A}_{\text{AC2}}^{\text{cf}}$ be a "canceling" intervention of depth 1, i.e., $d(C) = 1$. Then, for all $X \in C$, we have $\text{Ch}(X) = \{T\}$. Hence, $C$ is part of the parent of $T$, i.e., $C \subseteq \text{Pa}(T)$. Then, by definition, it is a constrained "canceling" intervention with constraining set $\kappa_0$ (the parents of the target is the free instance as only constraint), i.e., $(C, c) \in \mathcal{A}_{\text{AC2}}^{\text{cf}}[\kappa_0]$ and $\kappa_0 = (\text{Pa}(T), \emptyset, \emptyset, \emptyset, \emptyset)$. Additionally, this constraining set is part of its CF-closure, i.e., $\kappa_0 \in \mathcal{CFX}^0(\{\kappa_0\}) \subseteq \mathcal{CFX}^*(\{\kappa_0\})$. Hence, the "canceling" intervention $(C, c)$ is identified with a constraining set of the CF-closure of $T$, i.e., $(C, c) \in \bigcup(...)$.*

**Induction.** *Let*
$$n \geq 1 \quad and \quad (C', c') \in \mathcal{A}_{\text{AC2}}^{\text{cf}} \text{ such that } d(C') = n + 1 \tag{1}$$

*$(C', c')$ is a "canceling" intervention of depth $n + 1$. By the induction hypothesis, we suppose that any "canceling" intervention with depth lower than or equal to $n$ is a constrained "canceling" intervention with constraining set in the CF-closure of the target. We aim to prove that there is a constraining set $\kappa'$ in the*

CF-closure of the target $\mathcal{CFX}^*(\{\kappa_0\})$ such that $(C', c')$ is a "canceling" intervention constrained by $\kappa'$, i.e., $(C', c') \in \mathcal{A}_{\text{AC2}}^{\text{cf}}[\kappa']$.

Let

$$W' \in \mathcal{W}(C', c') \qquad and \qquad \xi' = [C' \leftarrow c', W' \leftarrow w'^*] \tag{2}$$

be some witness for $(C', c')$ and the intervention that satisfy AC2 involving $(C', c')$ and $W'$, i.e., $(\mathcal{M}, u) \models [\xi'](\neg T)$.

Let

$$S' = \{X \in C' | d(X) = n + 1\} \tag{3}$$

be the set of variables of $C'$ of depth $n + 1$ and let

$$S = \{X \in \text{Ch}(S') \setminus C' \mid \exists Y \in S', \exists P \in \mathcal{P}_{\text{cf}}(Y; C', c', W') : X \in P\} \tag{4}$$

be the set of strict children of $S'$ that are on cf-active paths from $S'$ to $T$. $S$ is well defined because $(C', c') \in \mathcal{A}_{\text{AC2}}^{\text{cf}}$ by definition.

Let

$$C = (C' \setminus S') \cup S \tag{5}$$

be the set of variables obtained by replacing $S'$ by $S$ in $C'$.

Since variables in $S$ are children of variables with depth $n + 1$, the depth of $S$ is $n$. Since $C' \setminus S'$ is composed of variables of $C'$ (with depth lower than or equal to $n + 1$) without the variables of $S'$ (all those of depth $n + 1$), then $d(C' \setminus S') \leq n$, which implies that $\boxed{d(C) = n}$. Additionally, since the variables in $S$ are the children of variables in $S'$ that belong to counterfactual paths, we can note s their counterfactual values under intervention $\xi'$, i.e., $(\mathcal{M}, u) \models [\xi'](S = s)$. Let $c$ be the counterfactual values of $C$ under $\xi'$.

$$c \in \text{Dom}(C) \ such \ as \qquad c_{|S} = s \qquad c_{|C \setminus S} = c'_{|C \setminus S} \tag{6}$$

Notably, $c$ is the counterfactual value of $C$ under $\xi'$, i.e., $(\mathcal{M}, u) \models [\xi'](C = c)$.

All children of $S'$ that are not in $S$ are not on a cf-active path. Hence, for any path from one of these variables to $T$, there will be a "blocking variable," i.e., a variable with all children taking their actual values. This implies that all these variables have no impact on $T$ nor on any variables that themselves have an impact on $T$. Hence, the impact of $C' \leftarrow c'$ only passes through $C = c$. Hence, $\boxed{(\mathcal{M}, u) \models [C \leftarrow c, W' \leftarrow w'^*](\neg T)}$.

Hence, $(C, c)$ is a "canceling" intervention with $d(C) \leq n$. Hence, according to the induction hypothesis, there is a constraining set $\kappa = (I, W_0, R_C, r_C, R_W) \in \mathcal{CFX}^*(\{\kappa_0\})$ such that $(C, c) \in \mathcal{A}_{\text{AC2}}^{\text{cf}}[\kappa]$.

Let

$$W \in \mathcal{W}_{R_W, I \cup W_0}(C, c) \qquad and \qquad \xi = [C \leftarrow c, W \leftarrow w^*] \tag{7}$$

be a witness for $(C, c)$ under counstraint $\kappa$ and let $\xi$ be the intervention that cancels $T$ involving $C$, $c$ and $W$.

Now we analyze the backtracking step that creates a constraining set $\kappa' \in cfx(\kappa)$ such that $(C', c') \in \mathcal{A}_{\text{AC2}}^{\text{cf}}[\kappa']$.

Since $S \subseteq C$ and $S \neq \emptyset$, let the constraining set according to $cfx$ with $(C, c)$, $W$ and $S$ be

$$\kappa' = (I', W_0', R_C', r_C', R_W')$$
$$I' = \text{Pa}(S) \setminus C \qquad W_0' = \text{De}^-(I') \cap \text{An}^-(C) \tag{8}$$
$$R_C' = (C \setminus S) \qquad r_C' = c_{|R_C'} \qquad R_W' = W \cup \text{De}^{-\setminus C}(I')$$

We now aim to prove that $(C', c') \in \mathcal{A}_{\text{AC2}}^{\text{cf}}[\kappa']$. This encompasses two facts (A) there exists witness for $(C', c')$ constrained by $\kappa'$, i.e., $\mathcal{W}_{R_W', I' \cup W_0'}(C', c') \neq \emptyset$. (B) The pair $(C', c')$ is constrained by $\kappa'$, i.e., $R_C' \subseteq C' \subseteq R_C' \cup I'$ with $c'_{|R_C} = r_C'$. Before proving (A) and (B), we first prove the useful result $S' \subseteq I'$.

We know that $C'$ is a cf-active AC2 "canceling" intervention, which means that all the variables in $C'$ have a non-empty set of cf-active paths from $C$ to $T$. Therefore, this is also true for all variables in $S'$, which is a subset of $C'$. By definition of $S$, this implies that, despite $S$ being a (possibly strict) subset of $\mathrm{Ch}(S')$, all variables in $S'$ have at least one child in $S$. Hence, all variables in $S$ have at least one parent in $S'$, which implies that $S' \subseteq \mathrm{Pa}(S)$. Since $C$ is obtained by removing variables from $S'$, removing variables from $C$ in $\mathrm{Pa}(S)$ does not remove variables from $S'$. Hence, $\boxed{S' \subseteq I'}$.

We now proceed to proving (A) and (B).

(A) Let

$$W'' = R'_W \cup (W' \cap W'_0) \tag{9}$$

be our candidate as constrained witness for $(C', c')$ under constraining set $\kappa'$.

We now need to show that $W'' \in \mathcal{W}_{R'_W, I' \cup W'_0}(C', c')$ by proving $R'_W \subseteq W'' \subseteq R'_W \cup W'_0 \cup I'$, and $(\mathcal{M}, u) \models [C' \leftarrow c', W'' \leftarrow w''^*](\neg T)$.

By construction, it is immediate that $R'_W \subseteq W''$. Additionally, it is also immediate that $(W' \cap W'_0) \subseteq W'_0 \subseteq W'_0 \cup I'$. Hence $W'' = R'_W \cup (W'_0 \cap W') \subseteq R'_W \cup W'_0 \cup I'$. Hence $\boxed{R'_W \subseteq W'' \subseteq R'_W \cup W'_0 \cup I'}$.

We now proceed to proving $(\mathcal{M}, u) \models [C' \leftarrow c', W'' \leftarrow w''^*](\neg T)$. For easier notation, we will denote $W_i = (W' \cap W'_0)$ and $Z = \mathrm{De}^{-\backslash C}(I')$.

Recall that $W_i$ is the set of variables of $W'$ on paths from $I'$ to $C$, which include the paths from $S'$ to $C$ (since $S' \subseteq I'$), which constitute the paths from $C'$ to $C$. Hence, if we use $W_i$ instead of $W'$ in $\xi$, we still obtain $(C = c)$, i.e., $(\mathcal{M}, u) \models [C' \leftarrow c', W_i \leftarrow w_i^*](C = c)$.

By Lemma 4, we can suppose that $W \in \mathrm{De}(C)$. If we add an intervention on $W$, we do not change the values of $C$, i.e., $(\mathcal{M}, u) \models [C' \leftarrow c', W_i \leftarrow w_i^*, W \leftarrow w^*](C = c)$.

Since $Z \cap \mathrm{An}(C) = \emptyset$, we can also add it to the intervention and obtain $C = c$, i.e., $(\mathcal{M}, u) \models [C' \leftarrow c', W_i \leftarrow w_i^*, W \leftarrow w^*, Z \leftarrow z^*](C = c)$. Similarly, since $W_i$ and $Z$ are not in the descendant of $C$, they naturally have there actual values under $\xi$. Hence, we also have $(\mathcal{M}, u) \models [C \leftarrow c, W_i \leftarrow w_i^*, W \leftarrow w^*, Z \leftarrow z^*](\neg T)$. Since $C' \leftarrow c'$ implies $C = c$, $C \leftarrow c$ implies $\neg T$ and all residual effect of $C' \leftarrow c'$ are blocked by $Z$, we finally have $\boxed{(\mathcal{M}, u) \models [C' \leftarrow c', W'' \leftarrow w''^*](\neg T)}$.

We successfully proved (A), i.e., $\boxed{\mathcal{W}_{R'_W, I' \cup W'_0}(C', c') \neq \emptyset}$.

(B) We now proceed to proving that the pair $(C', c')$ is constrained by $\kappa'$, i.e., $R'_C \subseteq C' \subseteq R'_C \cup I'$ with $c'_{|R_C} = r'_C$.

Since $R'_C = C \setminus S$, and $C = (C' \setminus S') \cup S$ we have that $R'_C$ is the "non-forwarded" part of the $C'$, i.e., $R'_C = C' \setminus S'$, which naturally imply $R'_C \subseteq C'$. Since we just saw that $R'_C = C' \setminus S'$ and we proved earlier that $S' \subseteq I'$, it immediately follows that $C' \subseteq R'_C \cup I'$. Combining the two inclusions yields $\boxed{R'_C \subseteq C' \subseteq R'_C \cup I'}$.

By construction, $r'_C = c_{|R'_C}$ (equation 8) and $c_{|C \setminus S} = c'_{|C \setminus S}$ (equation 6). Since $R'_C = (C \setminus S)$, we have $\boxed{c'_{|R'_C} = r'_C}$.

These two steps prove (B) the pair $(C', c')$ is constrained by $\kappa'$, i.e., $R'_C \subseteq C' \subseteq R'_C \cup I'$ with $c'_{|R_C} = r'_C$.

As we both showed that (A) there exists witness for $(C', c')$ constrained by $\kappa'$, i.e., $\mathcal{W}_{R'_W, I' \cup W'_0}(C', c') \neq \emptyset$, and (B) the pair $(C', c')$ is constrained by $\kappa'$, i.e., $R'_C \subseteq C' \subseteq R'_C \cup I'$ with $c'_{|R_C} = r'_C$, we can conclude that $\boxed{(C', c') \in \mathcal{A}^{\mathrm{cf}}_{\mathrm{AC2}}[\kappa']}$.

Since $\kappa' \in cfx(\kappa)$ and $\kappa \in \mathcal{CFX}^*(\{\kappa_0\})$, we have $\boxed{\kappa' \in \mathcal{CFX}^*(\{\kappa_0\})}$.

To wrap up, we showed that if any $(C, c) \in \mathcal{A}^{\mathrm{cf}}_{\mathrm{AC2}}$ with $d(C) \leq n$ is a constrained "canceling" intervention with some constraining set $\kappa \in \mathcal{CFX}^*(\{\kappa_0\})$, then any "canceling" intervention $(C', c') \in \mathcal{A}^{\mathrm{cf}}_{\mathrm{AC2}}$ with $d(C') = n+1$

is a constrained "canceling" intervention with some constraining set $\kappa' \in \mathcal{CFX}^*(\{\kappa_0\})$, i.e. $(C', c') \in \mathcal{A}_{\text{AC2}}^{\text{cf}}[\kappa']$.

**Conclusion** *Since we proved the initialization and the induction part of our hypothesis, we can conclude that any "canceling" intervention $(C, c) \in \mathcal{A}_{\text{AC2}}^{\text{cf}}$ is a constrained "canceling" intervention with some constraining set $\kappa \in \mathcal{CFX}^*(\{\kappa_0\})$. Hence* $\boxed{\mathcal{A}_{\text{AC2}}^{\text{cf}} \subseteq \bigcup(...)}$. ∎

We now recall Theorem 1 as expressed in the main paper and in a more formal way, we then prove it by combining our results.

**Theorem 1** *(ISI Completeness)*

**Main content version.** *For any SCM, every HP-cause $C$ appears as the variable set of some "canceling" intervention generated by the exact ISI algorithm.*

**Formal version.** *An algorithm that returns the set of minimal sets of variables $C$ for all $(C, c)$ in $\bigcup_{\kappa \in \mathcal{CFX}^*(\{\kappa_0\})} \mathcal{A}_{\text{AC2}}^{\text{cf}}[\kappa]$ returns exactly the full set of HP-causes $\mathcal{C}$.*

**Proof** *By Lemma 5 every HP cause $C \in \mathcal{C}$ is CF-active, hence $(C, c) \in \mathcal{A}_{\text{AC2}}^{\text{cf}}$ for some $c$. By Lemma 6 every $(C, c) \in \mathcal{A}_{\text{AC2}}^{\text{cf}}$ is in $\bigcup_{\kappa \in \mathcal{CFX}^*(\{\kappa_0\})} \mathcal{A}_{\text{AC2}}^{\text{cf}}[\kappa]$. Additionally, by construction of CF-active AC2 constrained pairs, we imediatly have*

$$\bigcup_{\kappa \in \mathcal{CFX}^*(\{\kappa_0\})} \mathcal{A}_{\text{AC2}}^{\text{cf}}[\kappa] \subseteq \bigcup_{\kappa \in \mathcal{CFX}^*(\{\kappa_0\})} \mathcal{A}_{\text{AC2}}[\kappa]$$

*Assuming ISI is implemented properly and follows its objective of exploring the constraining sets yielded by $\mathcal{CFX}^*(\{\kappa_0\})$, and combining both lemmas with the previous inclusion yields the theorem.* ∎

### F.3 Proof of Theorem 2

This subsection introduces additional notions necessary to prove Theorem 2 and proceeds to prove the introduced results as well as Theorem 2 itself.

This subsection aims to prove that, in a Boolean model, all HP-causes can be identified by performing a recursive "causal backtracking" on the already identified causes, starting from the target's parents. As we saw at the beginning of this section, the idea of the proof is to perform an induction on the "depth" of the causes. We suppose that we have identified all causes up to a certain "depth" $n$ and we consider some cause $C'$ at depth $n + 1$, build a cause $C$ at depth $n$, and show that we can "causally backtrack" from $C$ to $C'$.

The main challenge of this proof lies in building $C$ such that "causal backtracking" identifies $C'$. Indeed, in the previous proof, $C$ was simply composed of the counterfactual children of $C'$. But in this subsection, we do not consider full "canceling interventions" but minimal ones. Hence, we need to construct $C$ much more carefully by replacing, in $C'$, only the variable with the "highest precedence" (Definition 20) by its counterfactual children. When doing this, we must also ensure that $C$ has a "lower depth". Hence, we use a super-increasing score (Definition 18) as our notion of "depth".

For the remainder of this section, we suppose Boolean domains for all variables. Hence, if some variable $X$ has an actual value $x^*$, we can refer to its only possible counterfactual value as $\neg x^*$ or simply $\neg *$ when the variable in question is not ambiguous (for instance, when referring to an intervention such as $[X \leftarrow \neg *]$).

**Definition 16** *(Topological order) A topological order on the nodes in $\mathcal{V}$ is a bijection $\sigma : \mathcal{V} \to \{1, \dots, |\mathcal{V}|\}$ such that for all $A, B \in \mathcal{V}$, if $A \in \text{Pa}(B)$ then $\sigma(A) < \sigma(B)$.*

**Remark 7** *In a DAG, there is always a topological ordering. It can be obtained, for instance, by running a Depth-First graph traversal and reversing the final result.*

**Definition 17** *(Super-increasing weights) We call super-increasing weights a set of positive weights $(w_1, \ldots, w_n)$ satisfying the super-increasing condition:*

$$\forall i \in \{1, \ldots, n\} : w_i > \sum_{j=i+1}^{n} w_j$$

*Notably, using $w_i = 2^{n-i}$ is a natural choice as $w_i = 1 + \sum_{j=i+1}^{n} w_j$.*

**Definition 18** *(Super-increasing score of a set) Given a DAG with node $\mathcal{V}$, a topological order $\sigma$ and some superincreasing weight $(w_i)_{i \in \sigma(\mathcal{V})}$, we call the super-increasing score of a set $S \subseteq \mathcal{V}$ the value:*

$$s(S) = \sum_{X \in S} w_{\sigma(X)}$$

**Proposition 8** *(Properties of the super-increasing score) Let $A, B \subseteq \mathcal{V}$ and $S \subseteq A$ such that $S \neq \emptyset$.*

**Monotony:** *If $A \subseteq B$ then $s(A) \leq s(B)$.*

**Strict forward ordering:** *$\forall A' \subseteq \mathrm{De}(A) : s(A') < s(A)$*

**Strict forward decrease:** *$\forall S' \subseteq \mathrm{De}(S) : s\big((A \setminus S) \cup S'\big) < s(A)$*

**Proof** *Monotonicity.*

*Let $A, B \subseteq \mathcal{V}$ such that $A \subseteq B$. Then*

$$0 \leq s(A \setminus B) = \sum_{X \in A \setminus B} w_{\sigma(X)} = \sum_{X \in A} w_{\sigma(X)} - \sum_{X \in B} w_{\sigma(X)} = s(A) - s(B)$$

*Hence, we have $s(A) \leq s(B)$.* ∎

**Proof** *Strict forward ordering. Let $S \subseteq \mathcal{V}$, $S' \subseteq \mathrm{De}(S)$, and $i^* = \min_{X \in S}\big(\sigma(X)\big)$. Let $X^* \in S$ such that $\sigma(X^*) = i^*$, i.e., $X^*$ is the element of $S$ with the smallest topological index. Notably, $X^* \notin S'$, otherwise there would be some node in $S$ that would be its ancestor and would have a lower topological index.*

*Because $S' \subseteq \mathrm{De}(S)$ for any $X \in S'$, we have $\sigma(X) > \sigma(X^*) = i^*$. Hence, the following inequality holds: $s(S') = \sum_{X \in S'} w_{\sigma(X)} < w_{i^*} < s(S)$. Hence $s(S') < s(S)$.* ∎

**Proof** *Strict forward decrease. Let $A \subseteq \mathcal{V}$ and $S \subseteq A$ such that $S \neq \emptyset$.*

*If we remove $S$ from $A$, we remove $s(S)$ from $s(A)$. When we add $S'$ to $A \setminus S$, we add at most $s(S')$ to $s(A \setminus S)$. Hence, the following inequality: $s\big((A \setminus S) \cup S'\big) \leq s(A) - s(S) + s(S')$. Using the strict forward ordering property, we can conclude $s\big((A \setminus S) \cup S'\big) < s(A)$.* ∎

**Definition 19** *(Distance to a descendant) We call the distance between a variable $X \in \mathcal{V}$ and one of its descendants $Y \in \mathrm{De}(X)$ the size $dist(X, Y)$ of the smallest path from $X$ to $Y$. We extend this definition to the distance between $X$ and a subset of its descendants $S \subseteq \mathrm{De}(X)$ as the smallest distance between $X$ and any $Y \in S$.*

**Definition 20** *(Highest precedence element of a set) The highest precedence element of a set is the element that is the furthest from its shared descendant with any other element of the set.*

*Let $S \subseteq \mathcal{V}$, we call the highest precedence element of $S$:*

$$X_{max} = \underset{X \in S}{argmax}\Big[\underset{Y \in S}{\max}[dist\big(X, \mathrm{De}(Y) \cap \mathrm{De}(X)\big)]\Big]$$

**Proposition 9** *(Property of the highest precedence element) Given a set of variables $S \subseteq \mathcal{V}$ and its highest precedence element $X_{max}$, this property states that for any child $A \in \text{Ch}(X_{max})$, and any other variables $Y \in S$, either $A$ is a child of $Y$ or $A$ is not a descendant of $Y$. Formally:*

$$\forall A \in \text{Ch}(X_{max}), \ \forall Y \in S : A \in \text{Ch}(Y) \vee A \notin \text{De}(Y)$$

**Proof** *If $A$ is a descendant of $Y$, then the distance between $Y$ and $A$ is smaller than the distance between $X$ and $A$, which is 1. So $A \in \text{Ch}(Y)$.* ∎

We now introduce the most important lemma for proving Theorem 2. Most of this subsection's work lies in proving the following lemma.

**Lemma 10** *("Range" of $\mathcal{CFX}_b^*(\{\kappa_0\})$) All HP-causes $C \in \mathcal{C}$ are constrained HP-causes $C \in \mathcal{C}[\kappa]$ with constraining set $\kappa \in \mathcal{CFX}_b^*(\{\kappa_0\})$.*

$$\mathcal{C} \subseteq \bigcup_{\kappa \in \mathcal{CFX}_b^*(\{\kappa_0\})} \mathcal{C}[\kappa]$$

**Proof** *We proceed by induction over the super-increasing score of an HP-cause $n = s(C)$ (see Def. 18) for $C \in \mathcal{C}$. For easier notation, we denote $\bigcup(...) \equiv \bigcup_{\kappa \in \mathcal{CFX}_b^*(\{\kappa_0\})} \mathcal{C}[\kappa]$. We aim to prove that $\mathcal{C} \subseteq \bigcup(...)$.*

**Initialization.** *Let $n_0$ be the smallest possible super-increasing score for an HP-cause in $(\mathcal{M}, u)$. Let $C \in \mathcal{C}$ be an HP-cause of super-increasing score $n_0$, i.e., $s(C) = n_0$.*

*We now prove that $C$ is part of the parent of the target. By way of contradiction, suppose some variables $S \subseteq C$ are not part of the parents of $T$. Since $C \in \mathcal{C}$, there is a witness $W$ for which $T$ is canceled. Let*

$$A = \{X \in \text{Pa}(T) : (\mathcal{M}, u) \models [C \leftarrow \neg c^*, W \leftarrow w^*](X = \neg x^*)\}$$

*be the parents of $T$ that "flip" under the intervention that "flips" $T$.*

*We now that when $A$ take their counterfactual value and $\text{Pa}(T) \setminus A$ take their actual value (what happens under $[C \leftarrow \neg c^*, W \leftarrow w^*]$), $T$ flips. Since the value of $T$ is fully determined by the values of its parents only, we have $(\mathcal{M}, u) \models [A \leftarrow \neg a^*, \text{Pa}(T) \setminus A \leftarrow \neg *](\neg T)$.*

*Then, $A$ is a "canceling" intervention. Hence, either it is minimal, satisfies AC3, and is an HP-cause, or there is a minimal HP-cause included in it. Either way, $\exists A_C \subseteq A$ such that $A_C \in \mathcal{C}$.*

*Since the intervention $C$ can only affect its descendant, we have $A \subseteq \text{Pa}(T)$. Additionally, since $S$ are variables in $C$ but not in $\text{Pa}(T)$, the variables in $A$ that are not in $C$ are necessarilly descendant of $S$, i.e., $A \setminus C \subseteq \text{De}(S)$. We can rewrite this as $A = (C \setminus S) \cup S'$ with $S' \subseteq \text{De}(S)$. Hence, by strict forward decrease (Prop. 8), we have $s(A) < s(C)$. Then, by monotony (Prop. 8), we have $s(A_C) \leq s(A)$. Finally, we have $s(A_C) < s(C)$, which contradict our hypothesis. We just proved by contradiction that $\boxed{C \subseteq \text{Pa}(T)}$. By Lemma 4, we can say that there exists a witness in its descendant, which is necessarilly in the parents of $T$ too, i.e., $\exists W \in \mathcal{W}(C, \neg c^*)$ such that $\boxed{W \subseteq \text{Pa}(T)}$.*

*We can conclude that $C$ is a constrained HP-cause with constraining set $\kappa_0$ (the parents of the target is the free instance as only constraint), i.e., $C \in \mathcal{C}[\kappa_0]$ and $\kappa_0 = (\text{Pa}(T), \emptyset, \emptyset, \emptyset, \emptyset)$. Additionally, this constraining set is part of its boolean CF-closure, i.e., $\kappa_0 \in \mathcal{CFX}_b^0(\{\kappa_0\}) \subseteq \mathcal{CFX}_b^*(\{\kappa_0\})$. Hence, the HP-cause $C$ is identified with a constraining set of the boolean CF-closure of $T$, i.e., $C \in \bigcup(...)$.*

**Induction.** *Let*

$$n \geq n_0 + 1 \qquad and \qquad C' \in \mathcal{C} : s(C') = n \tag{10}$$

$C'$ is an HP-cause of *super-increasing score* $n$. By the induction hypothesis, we suppose that any HP-cause with *super-increasing score* strictly lower than $n$ is a constrained HP-cause with constraining set in the boolean CF-closure of the target. We aim to prove that there is a constraining set $\kappa'$ in the boolean CF-closure of the target $\mathcal{CFX}_b^*(\{\kappa_0\})$ such that $C'$ is an HP-cause constrained by $\kappa'$, i.e., $C' \in \mathcal{C}[\kappa']$.

Let

$$W' \in \mathcal{W}(C', \neg c'^*) \qquad and \qquad \xi' = [C' \leftarrow \neg c'^*, W' \leftarrow w'^*] \tag{11}$$

be some witness for $C'$ and the intervention that satisfy AC2 involving $C'$ and $W'$, i.e., $(\mathcal{M}, u) \models [\xi'](\neg T)$.

Let $X'$ be highest precedence element of $C'$ (see Def 20):

$$X' = \underset{X \in C'}{argmax} \Big[ \underset{Y \in C' \setminus \{X\}}{max} [dist\big(X, \mathrm{De}(Y) \cap \mathrm{De}(X)\big)] \Big] \tag{12}$$

Let

$$\widetilde{S} = \{X \in \mathrm{Ch}(X') \setminus C' \mid (\mathcal{M}, u) \models [\xi'](X = \neg x^*)\} \tag{13}$$

be the set of strict children of $X'$ that take counterfactual values under $\xi'$.

Let

$$\widetilde{C} = (C' \setminus \{X'\}) \cup \widetilde{S} \tag{14}$$

be the set of variables obtained by replacing $X'$ its counterfactual children $\widetilde{S}$ in $C'$.

All changes to the values of $\mathcal{V}$ due to $[X' \leftarrow \neg x'^*]$ are mitigated by its children. Hence, if we set by intervention the values of all children of $X'$ to the value they would have by $[X' \leftarrow \neg x'^*]$, we obtain the same effect, including $\neg T$. Hence, by definition of $\widetilde{C}$, we have $\boxed{(\mathcal{M}, u) \models [\widetilde{C} \leftarrow \neg \widetilde{c}^*, W' \leftarrow w'^*, \mathrm{Ch}(X') \setminus \widetilde{C} \leftarrow *](\neg T)}$.

Additionally, given some "canceling" intervention, there is a subset of it that is an HP-cause (if either is an HP-cause or has a subset that is by AC3). Hence, there is an HP-cause $\boxed{C \in \mathcal{C}}$ such that $\boxed{C \subseteq \widetilde{C}}$. Finally, by the strict forward decrease of the super-increasing score (see Prop 8), we also have $s(C) < s(C') = n + 1$. Hence, we have $\boxed{s(C) \leq n}$.

By the induction hypothesis, there is a constraining set $\kappa = (I, W_0, R_C, r_C, R_W) \in \mathcal{CFX}_b^*(\{\kappa_0\})$ such that $C \in \mathcal{C}[\kappa]$.

Let

$$W \in \mathcal{W}_{R_W, I \cup W_0}(C, \neg c^*) \qquad and \qquad \xi = [C \leftarrow \neg c^*, W \leftarrow w^*] \tag{15}$$

be a witness for $C$ under constraint $\kappa$ and let $\xi$ be the intervention that cancels $T$ involving $C$ and $W$.

Now we analyze the backtracking step that creates a constraining set $\kappa' \in cfx_b(\kappa)$ such that $C' \in \mathcal{C}[\kappa']$.

Let $S$ be

$$S = \widetilde{S} \cap C \tag{16}$$

the set of variables of $C$ that comes from $\widetilde{S}$. Then by construction $\boxed{S \subseteq C}$.

Let's assume by way of contradiction that $\widetilde{S} \cap C = \emptyset$. Recall that $C$ is the minimal "canceling" intervention based on $\widetilde{C} = (C' \setminus \{X'\}) \cup \widetilde{S}$. Hence, if there is no intersection between $C$ and $\widetilde{S}$, then $C \subseteq (C' \setminus \{X'\})$, i.e., $C \subsetneq C'$. Since $C'$ and $C$ are HP-cause, this contradicts our hypothesis. Hence, $\widetilde{S} \cap C \neq \emptyset$. Hence, $\boxed{S \neq \emptyset}$.

Since $S \subseteq C$ and $S \neq \emptyset$, let the constraining set according to $cfx_b$ with $C$, $W$ and $S$ be

$$\begin{aligned} \kappa' &= (I', W_0', R_C', r_C', R_W') \\ I' &= \mathrm{Pa}(S) \setminus C \qquad W_0' = \mathrm{De}^-(I') \cap \mathrm{An}^-(C) \\ R_C' &= (C \setminus S) \qquad r_C' = \neg c^*_{|R_C'} \qquad R_W' = W \cup \mathrm{De}^{-\setminus C}(I') \end{aligned} \tag{17}$$

We now aim to prove that $C' \in \mathcal{C}[\kappa']$. This encompasses two facts (A) there exists a witness for $C'$ constrained by $\kappa'$, i.e., $\mathcal{W}_{R'_W, I' \cup W'_0}(C', \neg c'^*) \neq \emptyset$. (B) The HP-cause $C'$ is constrained by $\kappa$, i.e., $R'_C \subseteq C' \subseteq R'_C \cup I'$.

We now proceed to proving (A) and (B).

(A) Let

$$W'' = R'_W \cup (W' \cap W'_0) \tag{18}$$

be our candidate as constrained witness for $C'$ under constraining set $\kappa'$.

We now need to show that $W'' \in \mathcal{W}_{R'_W, I' \cup W'_0}(C', \neg c'^*)$ by proving $R'_W \subseteq W'' \subseteq R'_W \cup W'_0 \cup I'$, and $(\mathcal{M}, u) \models [C' \leftarrow \neg c'^*, W'' \leftarrow w''^*](\neg T)$.

By construction, it is immediate that $R'_W \subseteq W''$. Additionally, it is also immediate that $(W' \cap W'_0) \subseteq W'_0$. Hence $W'' = R'_W \cup (W'_0 \cap W') \subseteq R'_W \cup W'_0 \cup I'$. Hence $\boxed{R'_W \subseteq W'' \subseteq R'_W \cup W'_0 \cup I'}$.

We now proceed to proving $(\mathcal{M}, u) \models [C' \leftarrow c', W'' \leftarrow w''^*](\neg T)$. For easier notation, we will denote $W_i = (W' \cap W'_0)$ and $Z = \mathrm{De}^{-\setminus C}(I')$.

Since $\xi'$ "flips" $C$, i.e., $(\mathcal{M}, u) \models [C' \leftarrow \neg c'^*, W' \leftarrow w'^*](C = \neg c^*)$, and $W' \setminus W_i$ has no causal effect on $C$, i.e., $\mathrm{De}^+(W' \setminus W_i) \cap \mathrm{An}^+(C) = \emptyset$, we can use $W_i$ instead of $W'$ to "flip" $C$, i.e. $(\mathcal{M}, u) \models [C' \leftarrow \neg c'^*, W_i \leftarrow w_i^*](C = \neg c^*)$.

By Lemma 4, we can suppose that $W \in \mathrm{De}(C)$. If we add an intervention on $W$, we do not change the values of $C$, i.e., $(\mathcal{M}, u) \models [C' \leftarrow \neg c'^*, W_i \leftarrow w_i^*, W \leftarrow w^*](C = \neg c^*)$.

Since $Z \cap \mathrm{An}(C) = \emptyset$, we can also add it to the intervention and obtain $C = \neg c^*$, i.e., $(\mathcal{M}, u) \models [C' \leftarrow \neg c'^*, W_i \leftarrow w_i^*, W \leftarrow w^*, Z \leftarrow z^*](C = \neg c^*)$. Similarly, since $W_i$ and $Z$ are not in the descendant of $C$, they naturally have their actual values under $\xi$. Hence, we also have $(\mathcal{M}, u) \models [C \leftarrow \neg c, W_i \leftarrow w_i^*, W \leftarrow w^*, Z \leftarrow z^*](\neg T)$. Similarly to what we saw in the proof of Th. 1, we have that $C' \leftarrow \neg c'^*$ implies $C = \neg c^*$, that $C \leftarrow \neg c^*$ implies $\neg T$ and all residual effects of $C' \leftarrow \neg c'^*$ is blocked by $Z$. Hence, we can conclude that $\boxed{(\mathcal{M}, u) \models [C' \leftarrow \neg c'^*, W'' \leftarrow w''^*](\neg T)}$.

We successfully proved (A), i.e., $\boxed{\mathcal{W}_{R'_W, I' \cup W'_0}(C', \neg c'^*) \neq \emptyset}$.

(B) We now proceed to proving that the HP-cause $C'$ is constrained by $\kappa'$, i.e., $R'_C \subseteq C' \subseteq R'_C \cup I'$.

$C$ is a subset of $\widetilde{C}$ and can be partitioned into variables that come from $\widetilde{S}$ (those in $S$ according to equation 16) and variables that come from $C' \setminus \{X'\}$. Indeed, variables from $\widetilde{S}$ are not in $C'$ by construction (equation 13). Hence, it is immediate that $C \setminus S \subseteq C' \setminus \{X'\} \subseteq C'$, hence $\boxed{R'_C \subseteq C'}$.

We will now prove that $C' \subseteq R'_C \cup I'$. We will show that $C' \cap (R'_C \cup I')$ is a "canceling" intervention. Since it is included in $C'$ and $C'$ is a minimal "canceling" intervention, this implies our objective.

According to equation 17, $R'_C \cup I' = (C \setminus S) \cup (\mathrm{Pa}(S) \setminus C)$. Hence, $C' \cap (R'_C \cup I') = \left(C' \cap (C \setminus S)\right) \cup \left(C' \cap (\mathrm{Pa}(S) \setminus C)\right)$. Additionally, by general set operations, we have $C' \cap (\mathrm{Pa}(S) \setminus C) = (\mathrm{Pa}(S) \cap (C')) \setminus C$. Since $C \setminus S \subseteq C'$ (indeed, we already saw that $R'_C \subseteq C'$), we have $C' \cap (R'_C \cup I') = C \setminus S \cup \left((\mathrm{Pa}(S) \cap C') \setminus C\right)$. Reorganizing variables in the union, we have $C \setminus S \cup \left((\mathrm{Pa}(S) \cap C') \setminus C\right) = C \setminus S \cup \left((\mathrm{Pa}(S) \cap C') \setminus S\right)$. Additionally, since $S \subseteq \widetilde{S}$ and $\widetilde{S} \cap C' = \emptyset$, we have $(\mathrm{Pa}(S) \cap C') \setminus S = \mathrm{Pa}(S) \cap C'$. Finally, we obtain $\boxed{C' \cap (R'_C \cup I') = C \setminus S \cup (\mathrm{Pa}(S) \cap C')}$.

We now show that $C \setminus S \cup (\mathrm{Pa}(S) \cap (C'))$ is a "canceling" intervention. By equation 16, we have $(\mathcal{M}, u) \models [C' \leftarrow \neg c'^*, W' \leftarrow w^*](C = \neg c^*)$. Additionally, by Prop. 9, all variables in $C'$ are either variables in the parent of $S$ or variables that do not belong to the ancestor of $S$ (and therefore have no causal effect on it). Hence, $(\mathcal{M}, u) \models [\mathrm{Pa}(S) \cap C' \leftarrow \neg *](S = \neg s^*)$. Similarly, since $C \setminus S \subseteq C'$, they also either belong to $\mathrm{Pa}(S) \cap C'$ or have no causal effect on $S$. Hence, $(\mathcal{M}, u) \models [C \setminus S \leftarrow \neg *, \mathrm{Pa}(S) \cap C' \leftarrow \neg *](S = \neg s^*)$. Since, $W \subseteq \mathrm{De}^-(S)$, we also have $(\mathcal{M}, u) \models [C \setminus S \leftarrow \neg *, \mathrm{Pa}(S) \cap C' \leftarrow \neg *, W \leftarrow w^*](S = \neg s^*)$. Since, $(\mathcal{M}, u) \models [C \leftarrow \neg c^*, W \leftarrow w^*](\neg T)$ and $C = C \setminus S \cup S$, we have $(\mathcal{M}, u) \models [C \setminus S \leftarrow \neg *, \mathrm{Pa}(S) \cap C' \leftarrow$

$\neg *, W \leftarrow w^*](\neg T)$. Hence, $(C \setminus S) \cup (\mathrm{Pa}(S) \cap C')$ is a "canceling" intervention. Hence, so is $C' \cap (R'_C \cup I')$. By the minimality of $C'$, we obtain $C' \cap (R'_C \cup I') = C'$, which imply $\boxed{C' \subseteq R'_C \cup I'}$.

Combining both inclusions, we have $\boxed{R'_C \subseteq C' \subseteq R'_C \cup I'}$.

As we both showed that (A) there exists a witness for $(C', \neg c'^*)$ constrained by $\kappa'$, and (B) the HP-cause $C'$ is constrained by $\kappa'$, we can conclude that $\boxed{C' \in \mathcal{C}[\kappa']}$.

Since $\kappa' \in cfx_b(\kappa)$ and $\kappa \in \mathcal{CFX}_b^*(\{\kappa_0\})$, we have $\boxed{\kappa' \in \mathcal{CFX}_b^*(\{\kappa_0\})}$.

To wrap up, we showed that if any $C \in \mathcal{C}$ with $d(C) \leq n$ is a constrained HP-cause with some constraining set $\kappa \in \mathcal{CFX}_b^*(\{\kappa_0\})$, then any HP-cause $(C', c') \in \mathcal{C}$ with $d(C') = n + 1$ is a constrained HP-cause with some constraining set $\kappa' \in \mathcal{CFX}_b^*(\{\kappa_0\})$, i.e. $C' \in \mathcal{C}[\kappa']$.

**Conclusion.** *Since we proved the initialization and the induction part of our hypothesis, we can conclude that any HP-cause $C \in \mathcal{C}$ is a constrained HP-cause with some constraining set $\kappa \in \mathcal{CFX}_b^*(\{\kappa_0\})$. Hence $\mathcal{C} \subseteq \bigcup(...)$.* ∎

We now recall Theorem 2 as expressed in the main paper, and in a more formal way, we then prove it by combining our results.

**Theorem 2** *(Boolean Completeness)*

**Main content version.** *When the SCM is Boolean, backtracking only from minimal "canceling" interventions (HP-causes) remains complete: every HP-cause is still discovered.*

**Formal version.** *Given a boolean SCM, an algorithm that returns the set of minimal sets of variables $C$ in $\bigcup_{\kappa \in \mathcal{CFX}_b^*(\{\kappa_0\})} \mathcal{C}[\kappa]$ returns exactly the full set of HP-causes $\mathcal{C}$.*

**Proof** *If we suppose that the ISI algorithm is implemented properly, then it explores the full set of constraining sets and retrieves all corresponding constrained HP-causes. Returning all the globally minimal ones yields all HP-causes.* ∎

