# OpenReview forum: "Searching for actual causes: Approximate algorithms with adjustable precision"
_TMLR — Under review for TMLR_

### Review · Reviewer_WNfg · 2026-06-04

**Summary Of Contributions:**

Computing actual causes can be incredibly inefficient. This paper introduces algorithmic approaches for identifying actual causes following the Halpern and Pearl definition. All approaches assume the existence of an oracle and a heuristic function for determining some kind of "distance" between an intervention and the effect cancellation. First, a beam search algorithm is introduced, followed by an iterative sub-instance decomposition that improves efficiency by leveraging the problem's graphical structure in a recursive manner. A confidence bounds approach enables finding actual causes in a similar manner for stochastic problems. The paper includes several sanity-check experiments to demonstrate correctness and evaluates the performance of algorithms on one example model in more detail.

**Additional Comments:**

**Suggestions for Improved Clarity**

- **C1.** The motivation talks a lot about XAI, not emphasizing single-instance-attributions, e.g., "Traditional XAI and causality aim to answer 'What factors contributed to this outcome?', i.e., general causation." I would say this is not generally true, as attribution methods, in particular, tend to account for the computational effects of single-instance cases. However, they do not focus on the "true" underlying real-world causality and are generally computational methods. Still, the introduction would benefit from some reframing by focusing on the causal and actual causality aspects.

- **C2.** In 4.1.1, what are the characteristics of a "promising" intervention?
- **C3.** In Algorithm 1, line 8: Why is "filterMinimality" here? I thought this filtering refers to "enforcing minimality during expansion" and should therefore happen earlier (around line 4).
- **C4.** Is the target information missing in Algorithm 1? I suppose it might be included through the oracle function, but I think it would be clearer if the target variable were made explicit in the inputs.
- **C5.** I find the examples in Figure 1 to be unnecessarily complicated and large. The purpose of having all those cases here, along with the detailed explanation, is not entirely clear to me, as it takes a lot of mental load to parse all of them, when maybe one example could be sufficient.
- **C6.** Please elaborate on how exactly "naive backtracking" would behave in 4.2.1.
- **C7.**  Before Section 4.2.3, the intuition behind the different variables introduced in 4.2.2. could be clarified better. I.e., what are variables, what are values, and what will they be used for?
- **C8.** What is $r_W^*$ (should be introduced) and why is it not part of $\kappa$?
- **C9.** How does "search" in Algorithm 2, line 6, work? Is that calling MBS?
- **C10.** Please elaborate on why the lower/upper confidence bound framework is applied here. Generally, we are not looking for a "best" intervention, but for those that change the outcome per the HP cause definition. I suppose this refers to the heuristic function? Please clarify.
- **C11.** In Appendix B, why is ISI sometimes slower than MBS? I thought one of the main motivations was increased efficiency?
- **C12.** The reference set is obtained via beam search with 𝑏 = -1. How exactly does this translate to the stochastic cases and how is the reference set obtained here?
- **C13.** While this is made explicit, I still dislike that all standard deviations in the figures are halved, as this greatly harms their intuitive understanding. If simply using the original values results in unreadable figures, a different type of visualization might be in order.

**Missing Definitions and Mistakes to be Fixed**

- **C14.**  "Intuitively, an actual cause is a sufficient and necessary condition for a target consequence." While this sentence talks about intuition, it is still factually false (actual causes can be neither sufficient nor necessary). This should be changed to account for this.
- **C15.** $u$ after Definition 1 is undefined.
- **C16.** The fact that $e$ is used for interventions is never introduced. More importantly, $E$ (Algorithm 1) is not defined.
- **C17.** Figure A.1 misses dotted lines in step 2, from the other gray nodes. Although I believe that omitting the children of the unselected might be better for clarity.
- **C18.** 4.2.3 Typo: "it supersets" to "its supersets"
- **C19.** The illustration in 4.2.3 uses witness $BH$. Should this not be $\neg BH$? I suppose this could be meant to only refer to the variable, not the value. But since the shorthand notation for variables is used here, it is unclear whether $BH$ should refer to the variable in general or to its state being true. In this case, it is particularly confusing (though to an extent it also applies to other illustrations).
- **C20.** 4.2.2. describes $R_W$ as "values", even though the upper case notation indicates variables. Given the original world, the specific values can be derived: $r_W^*$, but this variable for the value is also not defined.
- **C21.** The proof for Lemma 4 is incorrect. I assume the authors meant to write "De" instead of "An".
- **C22.** Appendix F: Define the reference set $R$.
- **C23.** Definition 8 uses a different order of subindices compared to Definition 6 for $\mathcal{W}$.
- **C24.** What is the tuple of four elements at the end of Definitions 8 and 9? Should that not be the 5-element tuple? Otherwise, what does it mean?
- **C25.** $V=AN(T)$ should be given as an assumption for this definition and not just stated as a fact.
- **C26.** Definition 13 is wrong. Either it is not strictly about the longest path, or the definition should not shortcut to 1 if $X$ is a parent of $T$, as there might be a longer path from $X$ to $T$ even if that is true.
- **C27.** $T$ is not introduced in Definitions 13 and 14.
- **C28.** In Definition 15, $\mathcal{A}_\text{AC2}^\text{cf}$ is used once as a function (of $\kappa$) and once as a set itself. This is inconsistent usage and ambiguous notation and must be changed.
- **C29.** $a$ is never introduced in 4.3.1.
- **C30.** DICE scores should range from 0 to 1, not 0 to 100. If these were normalized, this must be mentioned (Figure 3).
- **C31.** The comma before "5" in the second paragraph of 6.2 is misplaced.

**Other Suggestions**

- **C32.** Add space between the operator and operands in line 8, algorithm 1
- **C33.** Move Figure A.1 so it is not the first thing shown in the appendix.

**Questions**

Using a beam search approach implies that we do not aim to find all causes in general. Are some causes "better" or "more useful" than others? How does this connect to the choice of the heuristics function?

Further, what are the benefits and drawbacks of different heuristic functions? I also do not fully understand the comparison in Appendix E. Why do "random" and "constant" perform relatively well? These do not appear to be useful heuristics at all. The "Occam" heuristic, on the other hand, can easily be argued to be desirable, yet it performs very poorly. What are your thoughts on this topic?

**Audience:**

Yes

**Audience Explanation:**

The paper is well-motivated. While the Halpern and Pearl actual causation framework remains a topic of discussion, it remains an important subfield in the areas of causality, causal explanations, and explainable artificial intelligence. While most work in this area, to the best of my knowledge, tends to focus on theoretical results, soundness of definitions, and formal correctness and exhaustiveness, the computational angle is not often investigated in detail. To this end, this paper makes valuable contributions by trading off the addition of additional assumptions (heuristic function, specified causal graph) for feasible computability in both boolean and stochastic settings in a manner more efficient than previous research would allow.

**Broader Impact Concerns:**

I do not have any concerns regarding ethical implications.

**Claims And Evidence:**

No

**Claims Explanation:**

While I like the idea behind the paper, I find the current execution insufficient. Most importantly, there are too many mistakes and presentation issues at the moment. Further, the experimental limitation is limited, as it only focuses on a single graphical structure, raising the question how well the result would generalize to other problems. In more detail, these are my strengths and weaknesses.

**Strengths**

- The paper pursues a valuable goal and is well-motivated (see also my comment on the interest among TMLR's audience).
- The algorithms make intuitive sense and provide a nice, plausible trade-off between assumptions and efficiency.
- The confidence-bound-based approach for finding actual causes in stochastic problems is a helpful addition to deterministic problems and, in some sense, also softens the oracle assumption, as stochastic problems represent scenarios where an all-knowing oracle is not available.
- The sanity-check experiments provide convincing evidence of the correctness of the proposed algorithms.
- Limitations are discussed very openly and extensively. Major weaknesses are addressed, and vague directions for future research are outlined.

**Weaknesses**

- **W1 Errors, Presentation, and Clarity.** Presentation and clarity of the paper are currently very problematic. Most crucially, there are several mistakes and missing definitions. Further, several parts of the paper would greatly benefit from additional explanations. To keep this section of the review relatively short, I list some general problems below. A more extensive list with specific details can be found under "Additional Comments". While it is entirely possible that all these problems can be easily fixed (I did not find a problem that clearly invalidates the given claims), they make reviewing unreasonably difficult and time-consuming. Due to the large number of mistakes, **I stopped checking for further errors in the proofs after Definition 15**, as I do not think this is the reviewers' responsibility at this stage. I kindly request the author to do a thorough pass before giving this work up for re-review. Examples (not exhaustive):
  - Missing definitions and introductions of variables (e.g., C15, C16)
  - Errors in theory (e.g., C21, C26)
  - Ambiguous or inconsistent notation (e.g., C19, C28)
  - Clarity and presentation issues (e.g., C5, C6)

While I appreciate that limitations are made transparent, they still matter. Most importantly, the underlying assumptions and limitations of the experimental evaluation. In addition to what is mentioned in the paper, I have a few additional comments:

- **W2 Experimental Evaluation.** The main experimental evaluation only considers one specific SCM. While the sanity checks in the appendix provide evidence of general correctness, how well the other results for the SMK scenario generalize to other problems remains unclear. Thus, the evaluation of computational efficiency (time and calls) and the number of HP-causing events identified under different parameters is limited. Further, the ILP limitation to 7 attackers means that there is no comparison to ILP in the large variable regime, where ISI is supposed to shine.
- **W3 Oracle Assumption.** The oracle assumption means that a causal model (e.g., a fully-specified SCM) is required that is capable of providing counterfactual outputs. This is notoriously difficult, as the model must be able to infer all information about exogenous variables, or the system must be highly deterministic. The stated examples should be able to provide such an oracle (simulator, controllable physical system), but the general usability is still limited by this. On the other hand, softening this weakness by treating problems as stochastic means that oracles do not need to be "all-knowing".

- **W4 Heuristic Function.** The assumption that a heuristic function is required is substantial. This, as well, is acknowledged by the authors, but further discussion on how to obtain heuristic functions, as well as their different benefits and disadvantages, is touched upon in different parts of the paper and would benefit from a more elaborate, distinct discussion.

**Requested Changes:**

Weakness 3 is substantial, but it is not a reason for rejection to me. The other weaknesses would benefit from changes, which I will list sorted by descending order of importance:

- **W1 Errors, Presentation, and Clarity.** Fix all the clarity and presentation issues commented on under "additional comments". Further, make a general, thorough pass over the paper to catch other problems I might have missed in my first review round. Ensure that notation is clear, consistent, and matches the explanations provided in the text.
- **W2 Experimental Evaluation.** Run experiments on other SCMs, varying the type of functions and graph structures. For example, synthetic experiments could be used for randomly generating graphs with different settings (number of nodes, sparsity/density) and different (stochastic) functions. Scenarios that could, thus, be covered include (but are not limited to) graphs where some nodes have a large number of parents and other  graphical structures, such as "diamond" structures, appear. It would also help if the authors could finish the ILP experiments even for larger problems.
- **W4 Heuristic Function.** In different sections of the paper, the implications and possible implementations of the heuristic functions are addressed (4.1.1, 7.1, Appendix E). Combining and extending this discussion in a separate section (for example, in the appendix, maybe adding on top of Appendix E) would help clarify the implications and help with coming up with heuristic functions. Also see my questions in additional comments.

---

> ### Author Response · Authors · 2026-06-22
> **Response to the review - general message and response to weakness 1, 2 and 4.**
>
> We warmly thank Reviewer 1 for their thorough reading and, in particular, for their attention to detail. We apologize for the mistakes that remained in the manuscript: we proofread the paper several times but lacked an external reader to catch the errors we had become blind to. We understand this is not the reviewer's responsibility, and we are grateful for the time spent. We have done, and will continue to do, several additional proofreading passes for the next revisions.
>
> ## Weakness 1 — Errors, presentation, and clarity
>
> We addressed all 33 listed points (see the comment-by-comment responses in the dedicated comment) and will run further passes for the next revision to catch errors beyond those flagged.
>
> ## Weakness 2 — Experimental evaluation
>
> We are designing the additional experiments and will add them in a supplementary appendix. This is a longer process, so the results are not in the current revision; we will upload a new revision as soon as they are available. Following the reviewer's suggestion, we will test the algorithms on controlled synthetic graphs with varying structure (e.g., high-parent nodes, diamond structures) and sparsity levels, and we will extend the ILP comparison to larger problems.
>
> ## Weakness 4 — Heuristic function
>
> We consolidated the discussion of the heuristic. Appendix E now restates and centralizes the comparison of heuristic choices, and Section 7.1 and Appendix E cross-reference each other so the implications of the heuristic choice are discussed in one place rather than scattered across Sections 4.1.1, 7.1, and Appendix E.

---

> ### Author Response · Authors · 2026-06-22
> **Response to the review - comment-by-comment responses C1-C10.**
>
> - **C1.** We removed the claim that XAI targets general rather than actual causation (i.e., that attribution methods are not local), which was indeed too strong. We replaced it with two separate, weaker claims: (1) the bulk of causal ML concerns *general, type-level* causation, whereas explanation calls for *actual, token-level* causation, which is what users typically expect (Halpern, 2016; Hitchcock, 2007; Miller, 2019); and (2) feature-attribution methods such as LIME and SHAP score how much each input *influences the model's prediction* — an association-based signal — rather than the minimal difference-making facts that actual causation captures; as Miller (2019) argues, such a list of weighted contributions is not yet an explanation. We also dropped the "Usage in XAI" row from Table 1, which carried the over-strong framing.
> - **C2.** We now state what a "promising" intervention is: one whose heuristic value indicates it is close to canceling T. Section 4.1.1 makes this explicit and points to the domain-guided heuristic in Section 7.1 as a concrete example of how proximity to canceling is evaluated.
> - **C3.** There are in fact two distinct minimality filterings, and we now explain both. The first happens during beam expansion (line 4) and discards candidates whose counterfactual part is a superset of an already-identified cause, avoiding wasted evaluations. The second (line 8) is still necessary because the causes discovered within the *same* step must also be checked for minimality against each other — line 4 cannot do this, since those causes are not yet known when the beam is expanded. We added this explanation and point to step 2 of the Appendix A example.
> - **C4.** We made the target explicit in the algorithm head: the oracle input ϕ in Algorithm 1 is now annotated as including the target T.
> - **C5.** Each graph in Figure 1 illustrates one component of the "causal backtracking" formulas, and we substantially expanded Section 4.2.1 so that each panel is now tied to the specific formula component it motivates. We kept the examples minimal to the best of our ability and believe the formulas are central and non-trivial, so we do not see which graph could be removed without dropping an illustrated case. Could the reviewer indicate whether some formula components are self-explanatory and need no illustration, or whether some of these graphs are non-minimal for the point they make?
> - **C6.** We added an explicit description of naive backtracking in Section 4.2.1: it would search only within the parents of the identified cause. We then walk through the five scenarios in Figure 1 to show concretely where this fails and what must be added in each case — preserving the current witness (NOT-graph), fixing alternate paths as witnesses (XOR-graph), treating intermediate variables as optional witnesses (chain-graph), backtracking from a non-minimal canceling intervention (split-graph), and considering all non-empty subsets (OR-graph).
> - **C7.** We reworked Section 4.2.2 to introduce each quantity explicitly and state its role: I is the free search space, W0 the optional witnesses, R_C / r_C the forced counterfactual interventions and their values, and R_W the forced witnesses. We added a paragraph after Algorithm 2 mapping each formula component back to the Figure 1 panel that motivates it.
> - **C8.** We now introduce r_W*: it denotes the actual values of the forced witness R_W, fixed by the context via (M, u) ⊨ (R_W = r_W*). Because these values are determined by the context, they need not be specified in the subproblem κ — which is precisely why r_W* is not part of it. This clarification was added in Section 4.2.2.
> - **C9.** We clarified that "search" is a pluggable subroutine that returns canceling counterfactual interventions together with a witness; the Algorithm 2 head now states this. In practice it is instantiated with MBS, as noted in the Section 4 overview ("ISI can call MBS to solve subproblems").
> - **C10.** We added a paragraph in Section 4.3.1 explaining the role of the confidence bounds. LUCB is not solving a best-arm / ranking problem. For each intervention it resolves two thresholding-and-membership decisions: whether E[ϕ(e)] confidently falls below the canceling threshold a, and whether the intervention belongs among the b interventions with the smallest ψ. The confidence-bound framework is used because it enables *adaptive sampling*: interventions whose status is already clear are not resampled, and only the ambiguous borderline cases are.

---

> ### Author Response · Authors · 2026-06-22
> **Response to the review - comment-by-comment responses C10-C33.**
>
> - **C11.** We added a note in Appendix B. ISI's speed-up comes from exploiting graph structure; the sanity-check models are small and flat, so there is no structure to exploit and the variable counts are too small for the asymptotic complexity to matter. In that regime ISI's overhead (subproblem generation and subset enumeration) can make it comparable to, or slower than, MBS. This is consistent with our main claim, since these examples lie outside the large-variable, structured regime where ISI is designed to help.
> - **C12.** We clarified how the reference set is obtained in Section 5.3. It is computed once with exact ISI (b = −1, no step limit) on the Boolean deterministic model. Because the non-Boolean and stochastic variants share the same ground-truth causes as their Boolean counterpart, and we reuse the same contexts, the same reference set serves all three versions; no separate reference set is computed for the stochastic case.
> - **C13.** We changed the figure statistics so they no longer halve the standard deviation. For the DICE figures (Figures 3 and 4a–b), the lines now report the median over the 20 contexts and the error bars report the 25–75 quantiles (IQR), which is robust to the large context-to-context variance. For the binary accuracy figure (Figure 4c), we report the mean and omit the y-axis error bar, since accuracy is binary. The captions of Figures 3 and 4 were updated accordingly.
> - **C14.** Fixed. The intuitive description no longer claims that actual causes are sufficient and necessary conditions.
> - **C15.** u is now introduced explicitly after Definition 1.
> - **C16.** We introduced the use of e for interventions in Section 4.1.1 and removed any mentions of E.
> - **C17.** We did not change this point: we understood that the reviewer ultimately agreed that “omitting the children of the unselected might be better for clarity.”
> - **C18.** Fixed ("it supersets" → "its supersets").
> - **C19.** We set every use of the shorthand interventions notation in bold, so the variable is now clearly distinguished from its positive value.
> - **C20.** Corrected; R_W is no longer described as "values," and the definition is now consistent with the fix for C8 (r_W* introduced as the value associated with the variable set R_W).
> - **C21.** Fixed: the proof of Lemma 4 now uses "De" instead of "An."
> - **C22.** The reference set R is now defined at the start of Appendix F.
> - **C23.** Fixed: Definition 8 now uses the same subindex order as Definition 6 for W.
> - **C24.** Fixed: the tuple at the end of Definitions 8 and 9 is now the intended 5-element tuple.
> - **C25.** V = An(T) is now stated as an assumption at the start of Appendix F rather than asserted as a fact.
> - **C26.** Corrected: Definition 13 no longer shortcuts to 1 when X is a parent of T, since a longer path from X to T may exist; the proof of Lemma 6 was updated to match.
> - **C27.** T is now introduced in Definitions 13 and 14, via a general introduction of the variables that remain constant throughout the formal section.
> - **C28.** We replaced 𝒜_AC2(κ) by 𝒜_AC2[κ] and 𝒜_AC2^cf(κ) by 𝒜_AC2^cf[κ], so the notation expresses that κ *constrains* 𝒜_AC2 / 𝒜_AC2^cf rather than turning a set into an operator.
> - **C29.** a is now introduced in the Section 4.3 introduction and recalled in 4.3.1.
> - **C30.** We state in Section 5.3 that DICE scores are reported in %.
> - **C31.** Fixed: the misplaced comma before "5" in the second paragraph of 6.2 is corrected.
> - **C32.** Fixed: a space was added between the operator and operands in line 8 of Algorithm 1.
> - **C33.** Figure A.1 was moved so it is no longer the first item in the appendix.

---

> ### Author Response · Authors · 2026-06-22
> **Response to the review - Answers to questions and note on weakness 3.**
>
> ## Answers to questions
>
> **Question 1 (are some causes “better”/“more useful”, and how does this relate to the heuristic?).**
>
> This is an important question, which we deliberately kept out of scope, as it deserves a dedicated study.
>
> Our position is that cause *identification* and cause *selection* should be treated as two separate steps: first identify a full or large set of causes, then filter or rank them using external criteria adapted to the user and context. This decoupling is deliberate: merging both objectives into a single search is not always feasible, and, as the answer to Question 2 shows, the signal for finding causes efficiently and the signal for identifying *good* causes can actively oppose each other.
>
> Our approach is designed with this two-step process in mind: we aim for the full set of causes, and even when it is not fully reached, the output is a large set of valid causes ready to be filtered or ranked. This is why we do not optimize for the “best” cause during search: the beam size is a tractability parameter, not a selection criterion.
>
> It could still be valuable to account for cause quality during the search itself, for instance to make the approximate set more representative of the full one. As Question 2 explains, this is not straightforward, and we leave it to future work.
>
> **Question 2 (why do “random”/“constant” do relatively well, while “Occam” does poorly?).**
>
> The search heuristic ψ estimates proximity to *canceling* T, which is different from — and can even oppose — what makes a cause a good explanation. “Occam” favors interventions that change the system as little as possible: desirable as a measure of explanation quality, but an intervention that barely changes the system is, almost by construction, unlikely to cancel T. As a *search* signal it therefore points away from the goal, which is why it performs poorly (as does “sum-neg”). “Random” and “constant” carry no signal but are neutral, so they sit above the counter-aligned heuristics and below “sum-pos”, which is aligned with canceling in our conjunction-heavy model.
>
> In short, a heuristic helps only insofar as it correlates with proximity to canceling T, and this correlation is model-dependent, so there is no universal choice. The finer ordering among the non-aligned heuristics is noisy, and we leave a definitive ranking to future work. We have revised Appendix E to make this explicit.
>
> ## Note on Weakness 3 — Oracle assumption
>
> We appreciate the reviewer's understanding that Weakness 3 is not a reason for rejection, and we agree it remains a substantial limitation for real-world deployment; we have kept this framing in the paper. We would add, however, that the oracle assumption should be read relative to what HP causation itself requires. Any method grounded in the HP framework needs a fully specified SCM, including structural equations and a causal graph. Our oracle is strictly *less* demanding: it requires only the ability to evaluate counterfactual outcomes, with no need for structural equations, and when MBS is used without ISI, no causal graph is needed at all. In this sense the oracle is a relaxation of the full-SCM requirement rather than an additional burden. We added a remark to this effect in Section 7.1.

---

> > ### Comment · Reviewer_WNfg · 2026-06-25
> >
> > I thank the authors for their detailed response. I appreciate the effort invested in improving the manuscript and the response so far resolved all the points it addressed without raising new concerns.
> >
> > In particular, I like the changes in the introduction regarding XAI (C1), the additional explanations regarding the graphical examples (C5) and the simple but elegant boldface notation to distinguish variables from values (C19), which I am sure would have saved me some time and confusion when first reading the paper. The authors' note on the oracle assumption is also entirely fair.
> >
> > Thus, I only have a few minor comments at the moment:
> >
> > I) Thank you for clarifying my question regarding the different heuristic functions and the idea that the tasks of finding any vs good causes can be orthogonal. I believe the additional explanation in Appendix E is quite useful. I am only wondering whether this "orthogonality" could also be stated a bit more explicitly in the paper, as, while it can be derived from the information already given, is a sufficiently interesting result to be stated more explicitly as well in my opinion.
> >
> > II) On C5/C6: I appreciate the push-back, as it indicates that the authors do not intend to sacrifice paper quality over appeasing reviewers. With these added changes, I entirely agree that the graphs should stay. Previously, this read to me as "naive backtracking is flawed, here are several examples on why, and next is the algorithm that fixes it". Now, the bridge connecting the examples to the algorithm is much more elegant. Apart from that, I think the new text in 4.2.2 is the more important clarification. If the authors were inclined to dial back some of the additional changes in 4.2.1, I would not be opposed. In particular, repeating "in addition to searching within the parent of the previously identified cause" every single time. I leave this up to the authors. Only one small question: why is there no reference to Figure 1d in 4.2.2?
> >
> > III) On C17: I apologize for being unclear here. I meant that I would remove the expanded nodes in step 2, i.e., {$BH, ST$}, {$BH, BT$}, {$BH, SH$}. Currently, I find it inconsistent that these are drawn, but that the other expansions of {$BH$} that are also expansions of "active" nodes (e.g., {$\neg ST, BH$}) are not connected to {$BH$} using dotted lines. I think removing nodes is preferable over adding additional dotted lines to avoid clutter.
> >
> > IV) Please take a look at Figure 4 to ensure all numbers are readable, especially on the top right of Subfigure c.
> >
> > I am really happy with the changes made in this revision. I look forward to the additional experimental results and will take a look at the proofs I have not checked so far in the next revision to see whether they resolve my remaining concerns.

---

### Review · Reviewer_nirB · 2026-06-16

**Summary Of Contributions:**

A mixture of theoretical and algorithmic advances, such as the decomposition of the actual-cause identification into sub-problems, motivating the proposed algorithm.

**Audience:**

Yes

**Audience Explanation:**

XAI and causality are well established in the ML community.

**Broader Impact Concerns:**

Not applicable.

**Claims And Evidence:**

Yes

**Claims Explanation:**

Formal proofs of the theorems.

**Requested Changes:**

Somewhat limited novelty: MBS algorithm requires a heuristic, which shifts finding the HP cause to finding/building a good heuristic. In my opinion, a somewhat obvious way of approaching the problem. Maybe clarify this and better highlight the novelties of the paper.

Figure 5: Looks like ILP line stops after the first five measurements!? Is this a mistake or any deeper reason behind this. Please elaborate on this or fix it.

References:
Please be consistent and ensure that every reference is either accompanied by a DOI or a link where it can be downloaded

Minor:
- Missing space after the comma in section 6.2 ",and 10 attackers."

---

> ### Author Response · Authors · 2026-06-22
> **Response to the review**
>
> We thank Reviewer 2 for their feedback. Our responses:
>
> - Novelty / “the heuristic shifts the problem to building a good heuristic”: We added a paragraph in the introduction that explicitly separates what we claim as substantial novel contribution from what is an adaptation of existing work. We agree that requiring a heuristic shifts part of the difficulty; our contribution is not the heuristic itself but (1) the first practical algorithm targeting the *full* set of HP-causes, including non-Boolean and stochastic models, and (2) the decomposition result (Theorem 1) and the ISI algorithm it justifies, which reduce the search even under a fixed heuristic. We also note that, unlike prior solver-based methods, MBS does not require the structural equations.
>
> - Figure 5 (ILP line stops after five measurements). This is not a mistake: as noted in Section 5.2, the ILP comparison relies on the free version of the Gurobi solver, which limits us to at most 7 attackers. We are working on running the full ILP experiments and will include them in the next revision.
>
> - References. Every reference now consistently includes a DOI or a download link.
>
> - Minor. The missing space after the comma in Section 6.2 was added.

---

### Review · Reviewer_5vF9 · 2026-07-18

**Summary Of Contributions:**

The paper studies practical identification of Halpern–Pearl actual causes and proposes three complementary algorithms: Minimal Beam Search, which approximately searches over counterfactual interventions using an oracle and heuristic; Iterative Sub-Instance, which exploits causal graph structure to decompose the search and is supported by a completeness result; and an LUCB-based procedure that reduces sampling costs in stochastic models. The main strength is that the work targets full cause-set identification beyond deterministic Boolean SCMs. However, the novelty of MBS and LUCB is largely based on adapting established search and bandit techniques. At the same time, the empirical evaluation is restricted to one synthetic SCM family and its simple non-Boolean noisy variants. More importantly, the algorithms require a counterfactual oracle and a domain-specific heuristic. Unfortunately, the paper gives little guidance on how either of those components could be estimated/validated in realistic ML applications. This substantially limits the practical significance of the results.

**Audience:**

No

**Audience Explanation:**

Researchers working on actual causation, formal causal reasoning, or counterfactual search may find the ISI decomposition and the empirical comparison of approximate search strategies interesting. However, the practical relevance to the broader ML community is limited by the method’s dependence on both a counterfactual oracle and a problem-specific heuristic.

In much of the ML literature, an oracle is used only to generate synthetic ground truth or evaluate an algorithm under controlled conditions. Here, the oracle is an essential component of the proposed algorithm. Therefore, it must evaluate the outcome under arbitrary, potentially unseen, or unrealistic interventions. Such an oracle would generally require access to the true structural equations or a highly accurate simulator. The paper itself acknowledges that learning this counterfactual oracle from data is beyond current ML capabilities and that interventions may be infeasible in real systems.

The same issue applies to the heuristic. Performance depends substantially on a heuristic that estimates how close an intervention is to canceling the target, yet no general procedure is provided for constructing or estimating this heuristic from data. The heuristic used in the experiments is tailored to the logical structure of the synthetic SMK benchmark, and the appendix shows that alternative heuristics can produce widely different results. The authors explicitly acknowledge that there are no general guidelines for heuristic design.

Therefore, the paper does not fully solve practical actual-cause identification. Instead, it assumes access to two components that may be as difficult to obtain as the original causal explanation problem.

**Claims And Evidence:**

No

**Claims Explanation:**

1) The novelty is narrower than the presentation suggests. MBS is primarily an adaptation of standard beam search with pruning rules derived from HP minimality, while LUCB is an adaptation of an established adaptive-sampling procedure. These are reasonable engineering choices, but their algorithmic novelty is limited. The main substantive contribution appears to be the ISI decomposition and its completeness result. Therefore, claims such as the “first practical algorithms” for broad classes of SCMs and a major step toward practical XAI appear overstated relative to what is demonstrated.

2) **The experimental evaluation is the main weakness.** All main experiments use a single synthetic SCM family, with the non-Boolean and stochastic experiments obtained through relatively minor modifications of the same graph. The non-Boolean version only enlarges a few discrete domains, while the stochastic version independently flips structural outputs with one fixed noise level. These experiments do not demonstrate robustness across different graph structures, structural functions, domain sizes, noise mechanisms, or realistic applications. The authors themselves acknowledge that the benchmark structurally favors ISI because the smallest cause is always among the direct parents of the target. This largely explains its perfect smallest-cause accuracy and weakens the comparison against ILP.

3) Most comparisons are between MBS and the authors’ own ISI extension. The ILP baseline is evaluated only for the smallest-cause problem and only on small models because of the chosen Gurobi license. I understand that there is no competing baseline for full cause-set identification. However, this is often not a good enough reason to limit experimentation in scientific publications. The authors must identify metric, and (multiple) case studies where their proposed algorithm outperforms existing methods.

4) The stochastic claim is under-supported as well. The experiments demonstrate efficient estimation of a chosen cancellation probability, but not necessarily identification of stochastic HP causes. Only one threshold, noise level, sampling budget, and confidence configuration is examined. While I can see that LUCB reduces calls by at most 27%, with some degradation in cause-set quality, there is no systematic parameter-sensitivity analysis.

**Requested Changes:**

1. The main experiments currently use one SCM family and construct the non-Boolean and stochastic settings as modifications of the same model. The authors also acknowledge that this benchmark favors ISI because the smallest cause lies among the target’s direct parents. I recommend evaluating at least two or three additional synthetic SCM families with substantially different properties, such as randomly generated Boolean SCMs with mixed logical functions, multi-layer graphs with causes that require several hops from the target, and non-Boolean or stochastic SCMs with varying domain sizes, graph densities, cause sizes, and witness-set sizes. These experiments should include cases specifically designed to be unfavorable to ISI, such as causes that are not direct parents of the target and models containing many non-minimal canceling interventions.

2. The broader practical claims would be more convincing with two or three real-world or high-fidelity simulator-based evaluations. Suitable examples could include an industrial process-control simulator with injected faults, a cyber-security testbed with known attack sources and propagation paths, and a multi-agent or robotic system with controlled collision or failure events. Because true HP causes are difficult to establish from observational datasets alone, the authors should select applications where interventions can be executed in a simulator or where ground-truth fault injections are available. Even small but carefully validated case studies would provide substantially stronger evidence of practical relevance than further variants of the current synthetic benchmark.

3. If the authors cannot come up with approaches to estimate the oracle and heuristics, then the authors should evaluate the method when the oracle is approximate rather than exact. For example, oracle classification errors, biased probability estimates, limited intervention budgets, and model misspecification can all serve as parameters to experiment with oracle approximation. They should also compare several generic heuristics across all benchmark families and investigate whether a heuristic learned from sampled interventions can replace the hand-designed domain-specific heuristic.

4. For small SCMs, the complete cause set should be independently verified using exhaustive enumeration, SAT, or ILP rather than relying only on an exact configuration of ISI to generate the reference set. The current full-identification evaluation compares against a reference produced by the authors’ own ISI implementation, making the validation partly self-referential.

5. Please perform ablation studies to evaluate the effects of minimality-based pruning, beam ranking, witness-variable handling, causal sub-instance decomposition, causal backtracking, early stopping, maximum search depth, and LUCB allocation. For ISI, the authors should compare against MBS run on the full graph and MBS restricted only to the target’s ancestors or parents. For LUCB, comparisons should control for the total number of samples and report both cause-set quality and the frequency of incorrect cancellation decisions.

6. In addition to ablation studies, there must be robustness studies in the paper. The results should be evaluated across multiple noise levels, cancellation thresholds, confidence levels, sampling budgets, and maximum numbers of search steps. Robustness to causal-graph errors should also be studied because ISI assumes that the relevant graph structure is available. Useful perturbations include missing edges, additional spurious edges, incorrect parent sets, and partial graph availability (these are just examples). The authors should report how such errors affect cause recall, precision, DICE score, model-call complexity and/or any other relevant metric.

7. The manuscript states that “Reyd et al. (2025) use MBS to identify the cause of a flock of artificial agents colliding with an obstacle,” while the present submission describes MBS as a newly introduced algorithm. The authors must clarify if MBS is part of "Background" or part of their contribution.